# TOPOLOGICAL CAUSAL EFFECTS

**Kwangho Kim,**[*] **Hajin Lee**
Department of Statistics, Korea University
145 Anam-ro, Seongbuk-gu, Seoul 02841, Korea
`{kwanghk, hlee2745}@korea.ac.kr`

## ABSTRACT

Estimating causal effects is particularly challenging when outcomes arise in complex, non-Euclidean spaces, where conventional methods often fail to capture meaningful structural variation. We develop a framework for topological causal inference that defines treatment effects through differences in the topological structure of potential outcomes, summarized by power-weighted silhouette functions of persistence diagrams. We develop an efficient, doubly robust estimator in a fully nonparametric model, establish functional weak convergence, and construct a formal test of the null hypothesis of no topological effect. Empirical studies illustrate that the proposed method reliably quantifies topological treatment effects across diverse complex outcome types.

## 1 INTRODUCTION

Causal inference has become a central tool across many disciplines, providing a statistical framework for learning intervention effects beyond mere association. In the potential-outcome framework (Imbens & Rubin, 2015), causal effects are defined by contrasting counterfactual outcomes, i.e., what would have happened under alternative treatment assignments. As modern scientific outcomes become increasingly complex, however, standard causal estimands and methods that rely on Euclidean summaries can fail to detect meaningful intervention-induced changes in the underlying structure of the outcome. Despite this practical relevance, comparatively little methodological work targets causal inference for outcomes that are intrinsically non-Euclidean or high-dimensional, and unstructured.

In this work, we focus on settings where scientifically salient intervention effects are expressed through changes in the outcome's topological structure, rather than through shifts in simple Euclidean summaries. Such effects arise naturally in the biomedical sciences, where treatments can change macromolecular conformations or folding patterns (Kovacev-Nikolic et al., 2016; Cang & Wei, 2018; Axelrod & Gomez-Bombarelli, 2022), in neuroscience, where stimuli can reshape brain connectivity (Sizemore et al., 2019), and in signal processing and medical imaging, where the goal is to detect structural changes in dynamical systems or diagnostic images (Kim et al., 2018; Gholizadeh & Zadrozny, 2018).

To formalize and estimate such effects, we introduce a class of causal estimands that quantify intervention-induced changes in the topological structure of complex outcomes using tools from topological data analysis (TDA). TDA extracts robust, multi-scale geometric and topological descriptors from complex data (Carlsson, 2020). We build on persistent homology, which summarizes how topological features (for example, connected components, loops, and voids) appear and disappear as the resolution parameter varies (Chazal & Michel, 2021). While TDA has been used to improve predictive robustness under distribution shift and perturbations (e.g., Carrière et al., 2020; 2021; Kim et al., 2020), and has been incorporated as a regularizer in conventional average treatment effect pipelines (Farzam et al., 2025), to our knowledge there is no existing framework that (i) defines a causal estimand directly in terms of topological summaries and (ii) provides corresponding nonparametric estimation and inference. Our work fills this gap by defining a topology-aware causal estimand and developing a corresponding efficient, doubly robust estimator with valid inference.

**Illustrative example.** We illustrate the proposed methodology using a macromolecule dataset (Axelrod & Gomez-Bombarelli, 2022), shown in Figure 1 and revisited in Section 6. Each molecule is

---

[*]Corresponding author. Both authors contributed equally to this work.

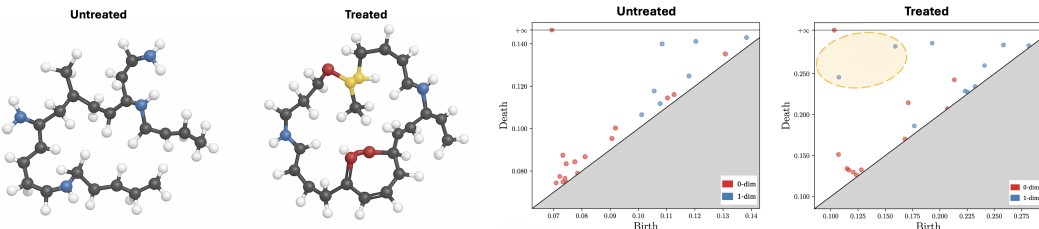

Figure 1: Left: Example of untreated vs. treated macromolecule structures. Right: Corresponding persistence diagrams, highlighting treatment-induced changes in the 1st-order homology features.

represented as a simplified graph, with graphs varying in size across samples. We consider a hypothetical chemical treatment that induces additional loop-like structure in the molecular geometry. Such changes can be difficult to detect using conventional Euclidean summaries, but they appear clearly in first-dimensional persistent homology, as evidenced by shifts in the corresponding persistence diagrams. We then convert these diagrams into functional summaries that are amenable to estimation and inference (Chazal et al., 2014; Bubenik et al., 2015). This perspective connects our approach to recent work on causal inference with functional outcomes (Ecker et al., 2024; Testa et al., 2025), a line of research that remains in the early stages of methodological development, particularly for fully nonparametric estimation and inference.

**Contributions.** We introduce a new class of *topological causal effects*, defined through intervention-induced changes in persistent homology summaries and, in particular, the expected contrast of power-weighted silhouette functions under the potential outcomes. This target is designed to capture structural effects that are invisible to scalar or Euclidean summaries, while retaining the stability properties of topological descriptors under perturbations. We develop an efficient, doubly robust, fully nonparametric estimator for the resulting functional estimand, obtain fast $\sqrt{n}$ rates under a standard product-rate condition on nuisance errors, and establish weak convergence, enabling valid and tractable inference in fully nonparametric settings. We also derive new stability bounds for weighted silhouettes under Wasserstein perturbations of persistence diagrams and, by combining these bounds with our weak convergence result, construct a formal test of the null hypothesis of no topological effect with asymptotically valid size and consistency. Together, our approach broadens the scope of causal inference by enabling rigorous analysis of structural effects in complex systems and supporting causal investigations in diverse modern data modalities, as illustrated in our empirical studies.

## 2 PRELIMINARIES: TOPOLOGICAL TOOLS FOR MACHINE LEARNING

This section provides a brief overview of foundational concepts in topological data analysis and introduces notation used throughout the paper. For further details, see Hatcher (2002); Edelsbrunner & Harer (2010); Chazal et al. (2016) as well as Appendix A.

**Simplicial complexes.** To represent the shape of complex data from a finite sample, we approximate the underlying object by a simplicial complex, a higher-dimensional generalization of a graph. Given a vertex set $V$, an (abstract) simplicial complex is a collection $K$ of finite subsets of $V$ such that if $\sigma \in K$ and $\tau \subseteq \sigma$, then $\tau \in K$. Each $\sigma \in K$ is called a simplex, its dimension is $\dim(\sigma) = |\sigma| - 1$, and the dimension of $K$ is the maximum simplex dimension among its simplices. In particular, a 1-dimensional simplicial complex corresponds to a graph (possibly with isolated vertices). In applications, simplicial complexes are typically constructed from observed point clouds using standard rules such as the Vietoris–Rips complex or the Čech complex.

**Persistent homology and diagrams.** A filtration is a nested family of simplicial complexes $\mathcal{F} = \{K(t) : t \in \mathbb{R}\}$ such that $t \leq t'$ implies $K(t) \subseteq K(t')$. Filtrations are often generated by a monotone filtration function $f : K \to \mathbb{R}$ satisfying $f(\tau) \leq f(\sigma)$ whenever $\tau \subseteq \sigma$, by setting $K(t) = f^{-1}((-\infty, t])$. Persistent homology summarizes how topological features emerge and disappear as the filtration parameter $t$ increases, separately for each homology degree $d = 0, 1, 2, \ldots$ (for example, $d = 0$ connected components, $d = 1$ loops, $d = 2$ voids). We write $\mathcal{H}_d\{K(t)\}$ for the

$d$-th homology group at level $t$, and represent each feature by its birth time $a$ and death time $b > a$; the resulting birth–death pairs $(a, b)$ form the persistence diagram. Formally, the $d$-th persistence diagram induced by $\mathcal{F}$ is denoted $\mathcal{D}_d(\mathcal{F})$ and is a multiset of points in

$$\mathbb{R}^{2+} := \{(a, b) \in (\mathbb{R} \cup \{\infty\})^2 : a < b\}.$$

When the filtration is clear from context, we write $\mathcal{D}_d$ for $\mathcal{D}_d(\mathcal{F})$, and we use $\mathcal{D}$ to denote a generic persistence diagram.

**Weighted silhouettes.** Because $\mathcal{D}$ is a multiset, it is often convenient to embed it into a function space. Given $\mathcal{D}$, define for each $p = (a_p, b_p) \in \mathcal{D}$ the tent function $\Lambda_p : \mathbb{R} \to \mathbb{R}$,

$$\Lambda_p(t) = \max\{0, \min\{t - a_p, b_p - t\}\}, \qquad t \in \mathbb{T}, \tag{1}$$

where $\mathbb{T} \subset \mathbb{R}$ is a fixed compact interval. The weighted silhouette is the normalized weighted average of these tents,

$$\phi(t; \mathcal{D}) = \frac{\sum_{p \in \mathcal{D}} w_p \, \Lambda_p(t)}{\sum_{p \in \mathcal{D}} w_p}, \qquad t \in \mathbb{T},$$

where $w_p \geq 0$ is the weight corresponding to point $p$, and $\sum_{p \in \mathcal{D}} w_p > 0$. A common choice is the power weight $w_p = (b_p - a_p)^r$ with $r > 0$, yielding the power-weighted silhouette

$$\phi(t; \mathcal{D}, r) = \frac{\sum_{p \in \mathcal{D}} (b_p - a_p)^r \, \Lambda_p(t)}{\sum_{p \in \mathcal{D}} (b_p - a_p)^r}, \qquad t \in \mathbb{T}. \tag{2}$$

Larger $r$ emphasizes longer-lived features, while smaller $r$ assigns relatively more weight to short-lived ones. Since $r$ only controls which features are emphasized and does not affect identification, estimation, or inference, we suppress $r$ in the notation when it is fixed.

The following lemma records a basic regularity property of silhouettes that will be used in Section 5.

**Lemma 2.1** (Lipschitz stability of the weighted silhouette). *For any $\delta > 0$,*

$$\sup_{|s-t| \leq \delta} \left| \phi(s; \mathcal{D}) - \phi(t; \mathcal{D}) \right| \leq \delta,$$

*and hence*

$$\mathbb{E}\left[ \sup_{|s-t| \leq \delta} \left| \phi(s; \mathcal{D}) - \phi(t; \mathcal{D}) \right| \right] \leq \delta.$$

**Choice of filtration.** Computing persistent-homology descriptors requires specifying a filtration $\mathcal{F} = \{K(t) : t \in \mathbb{R}\}$, typically obtained by choosing a simplicial (or cubical) complex $K$ together with a monotone filtration function $f : K \to \mathbb{R}$ and setting $K(t) = f^{-1}((-\infty, t])$. The appropriate construction depends on the data modality and the scientific notion of scale one wishes to probe. For point-cloud data, Vietoris–Rips and $\alpha$-filtrations are common choices; for grid-structured data such as digital images, sublevel or superlevel filtrations on cubical complexes are natural. For graph data, one can use clique filtrations, or the persistent homology transform (PHT) (Turner et al., 2014). These standard options allow our framework to accommodate a broad range of complex outcome types by pairing each modality with an appropriate filtration. Concrete definitions of several standard constructions (Vietoris–Rips, $\alpha$-, and cubical filtrations) are collected in Appendix A.

## 3 FRAMEWORK

Let $\{Z_i = (X_i, A_i, Y_i)\}_{i=1}^n$ denote an i.i.d. observed sample. Here, $A_i \in \mathcal{A} = \{0, 1\}$ is a binary treatment (intervention) indicator and $X_i \in \mathcal{X} \subseteq \mathbb{R}^l$ is an $l$-dimensional covariate vector. Let $\mathcal{F}_i$ denote the filtration of simplicial complexes constructed from $Y_i$, and let $\mathcal{F}_i^a$ denote the corresponding *potential (counterfactual) filtration* that would be constructed from the potential outcome $Y_i^a$ under treatment $A_i = a$. For each homology degree $d$, define the observed and potential persistence diagrams by $\mathcal{D}_{i,d} := \mathcal{D}_d(\mathcal{F}_i)$ and $\mathcal{D}_{i,d}^a := \mathcal{D}_d(\mathcal{F}_i^a)$. Also, let $\phi_{i,d}(t) := \phi\{t; \mathcal{D}_{i,d}\}$ and $\phi_{i,d}^a(t) := \phi\{t; \mathcal{D}_{i,d}^a\}$, for $t \in \mathbb{T}$, denote the corresponding power-weighted silhouettes in equation 2, where $\mathbb{T}$ is the domain of definition of $\phi$.

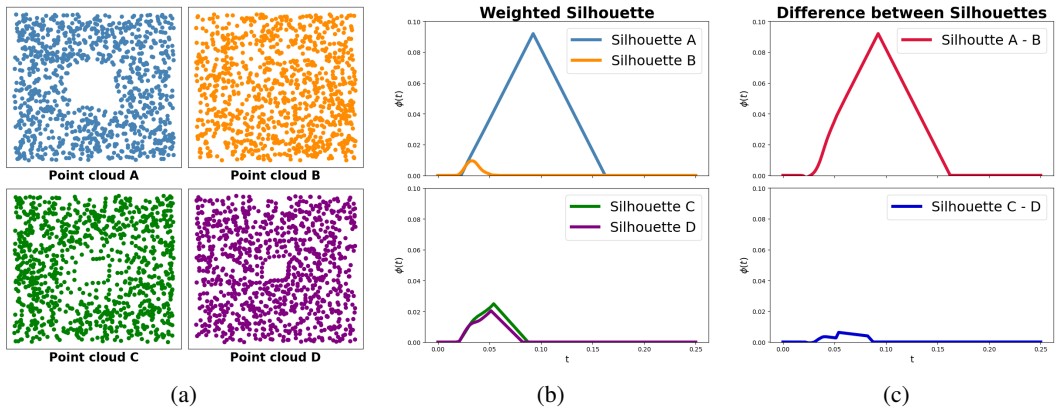

Figure 2: Silhouette functions revealing differences in 1-dimensional features in the ORBIT dataset (Adams et al., 2017). (a) Point clouds $A, B, C, D$; (b) their power-weighted silhouettes; (c) silhouette contrasts $\phi_A - \phi_B$ (top) and $\phi_C - \phi_D$ (bottom). Strong signals in $\phi_A - \phi_B$ indicate new 1-dimensional features in $A$ relative to $B$, whereas near-zero $\phi_C - \phi_D$ shows little change between $C$ and $D$.

Our target parameter of interest is the *topological average treatment effect* (TATE). For each fixed $d$, define the functional causal effect

$$\psi_d(t) := \mathbb{E}\left\{\phi_{i,d}^1(t) - \phi_{i,d}^0(t)\right\}, \qquad t \in \mathbb{T}. \tag{3}$$

Equivalently, for a chosen maximal homology degree $d_{\max}$, we may stack these components into the $\mathbb{R}^{d_{\max}+1}$-valued curve $\psi(t) = (\psi_0(t), \ldots, \psi_{d_{\max}}(t))$, $\forall t \in \mathbb{T}$, so that $\psi_d(t)$ is the $d$th coordinate of $\psi(t)$. Each function $\psi_d(t)$ captures the treatment-induced change in $d$-dimensional topological features at filtration scale $t$: $\psi_d(t) > 0$ indicates that, on average, treated units exhibit more or stronger $d$-dimensional features at scale $t$, while $\psi_d(t) < 0$ indicates attenuation under treatment. Thus, the mapping $t \mapsto \psi_d(t)$ is a functional causal effect in a Hilbert space, tracing how treatment alters topology across filtration scales.

There are several compelling advantages to interpreting equation 3 as treatment-induced topological effects. First, $\psi_d(t)$ is inherently scale-aware: the silhouette is indexed by the filtration parameter $t$, thereby retaining information about topology across geometric scales, thus allowing $\psi_d(t)$ to localize treatment-induced changes in topology across geometric scales. Second, it is robust to noise: power-weighted silhouettes downweight short-lived, potentially spurious features, so $\psi_d(t)$ emphasizes persistent structural differences that are more likely to be meaningful. Third, unlike raw persistence diagrams, $\psi_d(t)$ is vectorizable, in the sense that it lives in a separable Hilbert space, which facilitates theoretical analysis and integration into gradient-based machine learning pipelines. Fourth, $\psi_d(t)$ naturally fits within recent advances in functional causal inference. Collectively, $\psi_d(t)$ can be viewed as a topology-aware analogue of the average treatment effect, capturing how an intervention reshapes the $d$-dimensional homological structure of the underlying outcome across treatment groups.

One limitation is that silhouettes can obscure the exact number of changing homological features, since they aggregate multiple tent functions into a single weighted summary. In principle, however, the TATE can also be defined using individual persistence landscape functions across homology degrees up to a chosen level, which may provide finer topological resolution. Nevertheless, representing each diagram by a single silhouette curve offers a simple, interpretable summary of overall topological variation within a given homology degree, and the power parameter $r$ gives a transparent way to emphasize the most persistent features when desired.

Although one could apply standard average treatment effect estimators after vectorizing each persistence diagram into an ad hoc Euclidean feature vector, such heuristics generally lack a principled link to the underlying topological object. Our approach is fundamentally different: we define the causal estimand directly in terms of persistence-diagram geometry and use the silhouette as a functional embedding to obtain a well-posed target in a Hilbert space. This yields a coherent notion of a topological causal effect, along with doubly robust estimation and valid inference that are explicitly aligned with this target.

Figure 2 illustrates how the target parameter equation 3 captures structural variation. The point-cloud pairs $(A, B)$ and $(C, D)$ in Figure 2a represent potential outcomes under treatment and control. The treatment induces a loop in $A$ relative to $B$, producing a pronounced silhouette contrast in Figure 2c, whereas the contrast for $(C, D)$ is close to zero, indicating negligible structural change. The sign of the silhouette difference curve encodes the direction of topological change: positive values correspond to features that emerge under treatment, while negative values correspond to features that disappear relative to control.

For identification, we invoke the standard causal assumptions: (C1) *Consistency*, $\mathcal{F}_i = \mathcal{F}_i^a$ whenever $A_i = a$; (C2) *No unmeasured confounding*, $A_i \perp\!\!\!\perp \phi_{i,d}^a(t) \mid X_i$; and (C3) *Positivity*, $\mathbb{P}(A_i = a \mid X_i) > 0$ almost surely, for all $t \in \mathbb{T}$, $a \in \mathcal{A}$, and homology degree $d$. Despite the increased complexity of our target parameter, the identification conditions closely parallel those in standard causal inference settings (e.g., Imbens & Rubin, 2015). We assume that Assumptions (C1)–(C3) hold throughout the paper. Note that component-wise identification is sufficient for identifying the functional effect $\psi_d(t)$ at each filtration scale $t$. Then, for each $d$, the target $\psi_d(t)$ is identified as

$$\psi_d(t) = \mathbb{E}[\mathbb{E}\{\phi_{i,d}(t) \mid X_i, A_i = 1\} - \mathbb{E}\{\phi_{i,d}(t) \mid X_i, A_i = 0\}] \tag{4}$$

$$= \mathbb{E}\left\{ \frac{A_i\,\phi_{i,d}(t)}{\pi(X_i)} - \frac{(1 - A_i)\,\phi_{i,d}(t)}{1 - \pi(X_i)} \right\}. \tag{5}$$

The identifying expressions above motivate the use of plug-in and inverse probability weighting estimators, which will be discussed in greater detail in the following section.

**Remark 1** (Choice of $r$). *The TATE $\psi_d(t)$ depends on the power parameter $r$, which controls how strongly the silhouette emphasizes long-lived versus short-lived persistence pairs. We view $r$ as a problem-specific tuning parameter: practitioners may select it using domain knowledge (e.g., larger $r$ to upweight more persistent features), or by a simple data-driven criterion, such as checking the stability of $\widehat{\psi}_d$ over candidate values of $r$ (for example, via a validation step within a treatment arm or via cross-validation). In our experiments, the qualitative shape and statistical significance of the estimated effects remained stable across a modest range of $r$ values, suggesting limited sensitivity to its precise choice.*

## 4 ESTIMATION

In this section, we develop an estimation strategy for $\psi_d$ defined in equation 3. Hereafter, when no ambiguity arises, we omit the subscript indexing the $i$-th observation and write generic random variables in place of their sample realizations. For convenience, define the propensity score and the conditional silhouette regression functions by

$$\pi(x) := \mathbb{P}(A = 1 \mid X = x), \qquad \mu_a(t, x; d) := \mathbb{E}\{\phi_d(t) \mid X = x, A = a\}.$$

Let $\hat{\pi}$ and $\hat{\mu}_a$ denote estimators of $\pi$ and $\mu_a$, respectively. We write $\widehat{\mathbb{P}}$ for the empirical distribution used to fit the nuisance estimators $(\hat{\pi}, \hat{\mu}_0, \hat{\mu}_1)$, and we assume access to a separate empirical distribution $\mathbb{P}_n$, independent of $\widehat{\mathbb{P}}$, used to construct the final estimator.

Motivated by the identifying expressions in equation 4 and equation 5, we propose the plug-in (PI) regression estimator and the inverse-probability-weighting (IPW) estimator as

$$\widehat{\psi}_{\mathrm{PI},d}(t) = \mathbb{P}_n\{\hat{\mu}_1(t, X; d) - \hat{\mu}_0(t, X; d)\}, \tag{6}$$

$$\widehat{\psi}_{\mathrm{IPW},d}(t) = \mathbb{P}_n\left\{ \frac{A\,\phi_d(t)}{\hat{\pi}(X)} - \frac{(1 - A)\,\phi_d(t)}{1 - \hat{\pi}(X)} \right\}. \tag{7}$$

The estimators $\widehat{\psi}_{\mathrm{PI},d}$ and $\widehat{\psi}_{\mathrm{IPW},d}$ inherit their convergence rates from those of $\hat{\mu}_a$ and $\hat{\pi}$, respectively. In many observational studies, the IPW estimators can be appealing because modeling the treatment assignment mechanism is often more feasible than modeling the full conditional mean of a functional outcome (Imbens & Rubin, 2015). In our setting, $\widehat{\psi}_{\mathrm{IPW},d}$ can be particularly convenient because estimating the functional regression $\mu_a(\cdot, \cdot; d)$ may be substantially more challenging than estimating the scalar propensity score $\pi$.

A more efficient estimator can be constructed using semiparametric efficiency theory. Define the uncentered efficient influence function (EIF) by

$$\varphi_d(t, Z; \eta) := \mu_1(t, X; d) - \mu_0(t, X; d) + \left\{ \frac{A}{\pi(X)} - \frac{1-A}{1-\pi(X)} \right\} \{\phi_d(t) - \mu_A(t, X; d)\}, \quad (8)$$

where $\eta = \{\pi, \mu_0, \mu_1\}$ collects the nuisance functions. By construction, $\mathbb{E}\{\varphi_d(t, Z; \eta)\} = \psi_d(t)$ for each $t \in \mathbb{T}$. The influence-function representation yields an augmented IPW estimator,

$$\widehat{\psi}_{\mathrm{AIPW},d}(t) := \mathbb{P}_n\{\varphi_d(t, Z; \widehat{\eta})\} \equiv \mathbb{P}_n\{\widehat{\varphi}_d(t)\}, \quad (9)$$

where $\widehat{\eta} = \eta(\widehat{\mathbb{P}}) = \{\hat{\pi}, \hat{\mu}_0, \hat{\mu}_1\}$ is estimated using the nuisance-training sample $\widehat{\mathbb{P}}$. This construction yields the usual second-order remainder structure and the familiar double-robust behavior, allowing $\widehat{\psi}_{\mathrm{AIPW},d}$ to attain fast rates under weak nonparametric conditions on the nuisance estimators (Kennedy, 2016; 2024).

There are two standard approaches for establishing asymptotic properties of augmented inverse-probability-weighted estimators. One approach is to impose empirical-process conditions (e.g., Donsker or suitable entropy conditions) on the indexed class of influence functions $\{\varphi_d(t, \cdot; \eta) : t \in \mathbb{T}\}$, along with corresponding regularity of the nuisance estimators, but such conditions can be overly restrictive for modern, highly adaptive nuisance learning. An alternative is sample splitting, which allows arbitrarily complex nuisance estimators by separating nuisance fitting from the final debiasing step, thereby mitigating overfitting concerns. In principle, we remain agnostic about how the nuisance functions are learned. For clarity, however, we adopt a sample-splitting setup throughout: the nuisance estimators are fitted on $\widehat{\mathbb{P}}$, and the final estimator is constructed on an independent sample $\mathbb{P}_n$. (See Assumption (A2).)

**Remark 2** (Sample splitting). *Independent samples $\mathbb{P}_n$ and $\widehat{\mathbb{P}}$ can be obtained via random data splitting. Full-sample efficiency can be recovered via cross-fitting (e.g., Chernozhukov et al., 2018; Newey & Robins, 2018). For clarity of exposition, we work with a single split, though extending to multiple splits is straightforward (e.g., Kennedy, 2019; 2023; Kennedy et al., 2023).*

## 5 INFERENCE

Conditions guaranteeing desirable large-sample properties, such as $\sqrt{n}$-rate convergence and asymptotic normality, are well understood for average treatment effect estimation with scalar outcomes (e.g., Kennedy, 2016; 2024). In our setting, the outcome is a curve indexed by the filtration scale, so valid inference requires additional structure to control the behavior of the resulting stochastic process.

We begin with a simplified setting in which the treatment assignment mechanism is fully known, as in randomized experiments. In particular, we assume that the true propensity score $\pi$ is known almost surely. In this case, the IPW estimator in equation 7 is unbiased.

To obtain weak convergence of the entire estimated effect curve in $\ell^\infty(\mathbb{T})$, however, it is not enough to assume only pointwise second moments; one also needs a functional central limit theorem for the indexed empirical process $\{\xi_d(\cdot, t) : t \in \mathbb{T}\}$. This can be ensured, for example, by assuming that the indexed class is $\mathbb{P}$-Donsker (or by standard entropy-and-envelope conditions with suitable measurability). In our setting, these conditions are plausible because the silhouette $\phi_d(t)$ is Lipschitz in the index $t$ (see Remark 3 in Appendix C.2). The following theorem shows that, when $\pi$ is known, $\widehat{\psi}_{\mathrm{IPW},d}$ converges weakly to a Gaussian process in $\ell^\infty(\mathbb{T})$ under mild conditions.

**Theorem 5.1.** *For $t \in \mathbb{T}$, define $\xi_d(Z, t) := \left\{ \frac{A}{\pi(X)} - \frac{1-A}{1-\pi(X)} \right\} \phi_d(t)$, where $\pi$ is known. Assume that $\sup_{t \in \mathbb{T}} var(\xi_d(Z, t)) < \infty$ and that the class $\{\xi_d(\cdot, t) : t \in \mathbb{T}\}$ is $\mathbb{P}$-Donsker (or satisfies any other standard sufficient condition for a functional central limit theorem in $\ell^\infty(\mathbb{T})$). Then*

$$\sqrt{n}\{\widehat{\psi}_{\mathrm{IPW},d}(t) - \psi_d(t)\} \rightsquigarrow \mathbb{G}_d(t) \quad in \quad \ell^\infty(\mathbb{T}).$$

*where $\mathbb{G}_d$ is a mean-zero Gaussian process with covariance function $cov\{\mathbb{G}_d(s), \mathbb{G}_d(t)\} = cov(\xi_d(Z, s), \xi_d(Z, t)), \forall s, t \in \mathbb{T}$.*

In observational settings, it is often preferable to use the AIPW estimator defined in equation 9. Hereafter, we use the notation $\|\cdot\|_{\mathbb{P},q}$ to denote the $L_q(\mathbb{P})$-norm. To analyze the asymptotic properties, we introduce the following set of assumptions.

(A1) $\|1/\widehat{\pi}\|_\infty < \infty$ and $\|1/(1-\widehat{\pi})\|_\infty < \infty$.

(A2) *Sample splitting*: Nuisance estimates $\widehat{\eta}$ are fit on $\widehat{\mathbb{P}}$, and the final estimator is computed on an independent sample $\mathbb{P}_n$ ($\mathbb{P}_n \perp\!\!\!\perp \widehat{\mathbb{P}}$).

(A3) *Cross-sectional consistency*: $\|\widehat{\varphi}(t) - \varphi(t)\|_{\mathbb{P},2} = o_\mathbb{P}(1), \forall t \in \mathbb{T}$.

(A4) *Rate condition on nuisance estimation*:$\|\widehat{\pi}(X) - \pi(X)\|_{\mathbb{P},2} \left\{ \sum_{a\in\mathcal{A}} \|\hat{\mu}_a(t,X;d) - \mu_a(t,X;d)\|_{\mathbb{P},2} \right\} = o_\mathbb{P}(n^{-1/2}), \forall t \in \mathbb{T}$.

(A5) *Uniform consistency*: For all $x \in \mathcal{X}$, $\mu_a(t,x;d)$ is uniformly Lipschitz in $t$ on $\mathbb{T}$, and following conditions hold:

$$\sup_{t\in\mathcal{T},x\in\mathcal{X}} \left|\widehat{\mu}_a(t,x;d) - \mu_a(t,x;d)\right| = o_\mathbb{P}(1), \quad \sup_{x\in\mathcal{X}}\left|\widehat{\pi}(x) - \pi(x)\right| = o_\mathbb{P}(1).$$

(A5$'$) *Lipschitz modulus for $\hat{\mu}_a$*: For $\delta > 0$ and $a \in \mathcal{A}$, $\hat{\mu}_a(\cdot, X; d)$ satisfies $\mathbb{E}\left[\sup_{|s-t|\leq\delta} |\hat{\mu}_a(s,X;d) - \hat{\mu}_a(t,X;d)|\right] \leq L_{\hat{\mu}}\delta$, for some positive constant $L_{\hat{\mu}}$.

Assumptions (A1)–(A4) are standard in semiparametric causal inference (e.g., Kennedy, 2019; Kennedy et al., 2023; Kennedy, 2024). (A1) is a mild boundedness condition, while (A2) separates nuisance fitting from the final empirical average and thereby avoids empirical-process restrictions on the nuisance learners. (A3) enforces $L_2(\mathbb{P})$ consistency of the estimated influence function at each $t$, and (A4) ensures that the second-order von Mises remainder is $o_\mathbb{P}(n^{-1/2})$.

For the functional nuisance $\hat{\mu}_a(\cdot, \cdot; d)$, we consider two alternative regularity conditions. Assumption (A5$'$), adopted in Testa et al. (2025), imposes a Lipschitz modulus directly on the random sample paths of $\hat{\mu}_a$ by requiring an $L_1$ bound on $\sup_{|s-t|\leq\delta}|\hat{\mu}_a(s,X;d) - \hat{\mu}_a(t,X;d)|$. In contrast, Assumption (A5) requires only that the target regression $\mu_a(\cdot, \cdot; d)$ is uniformly Lipschitz in $t$ and that $\hat{\mu}_a$ converges uniformly to $\mu_a$. In fact, under a mild integrability condition such as $\mathbb{E}\{\sup_{t\in\mathbb{T}} |\hat{\mu}_a(t,X;d) - \mu_a(t,X;d)|\} = o(1)$, we have

$$\mathbb{E}\left[\sup_{|s-t|\leq\delta}\left|\hat{\mu}_a(s,X;d) - \hat{\mu}_a(t,X;d)\right|\right] \leq L_\mu\,\delta + o(1),$$

so $\hat{\mu}_a$ need not be smooth itself, provided it converges uniformly to a Lipschitz limit.

Depending on how $\hat{\mu}_a$ is learned, the regularity requirements in Assumptions (A5) and (A5$'$) can often be relaxed, and in some cases can be completely avoided, as illustrated by the following example.

**Example 5.1** (Functional linear smoother). *Consider a functional linear smoother for $\mu_a$ (e.g., Reiss et al., 2017): suppose that, for each $d$ and $a \in \{0,1\}$, the fitted regression admits the representation $\hat{\mu}_a(t,X;d) = \sum_{j=1}^n L_{d,j}(X)\,\phi_{j,d}(t)$, $t \in \mathbb{T}$, where $\phi_{j,d}(t) = \phi\{t; \mathcal{D}_{j,d}\}$ are the training silhouettes and $L_{d,j}(X)$ are (possibly data-adaptive) weights determined by the smoother, evaluated at $X$. Then, for any $\delta > 0$,*

$$\mathbb{E}\left[\sup_{|s-t|\leq\delta}\left|\hat{\mu}_a(s,X;d) - \hat{\mu}_a(t,X;d)\right|\right] \leq \mathbb{E}\left[\sum_{j=1}^n |L_{d,j}(X)| \sup_{|s-t|\leq\delta}\left|\phi_{j,d}(s) - \phi_{j,d}(t)\right|\right]$$

$$\leq \delta\,\mathbb{E}\left[\sum_{j=1}^n |L_{d,j}(X)|\right],$$

*where the last inequality follows from Lemma 2.1. In particular, if $\mathbb{E}\{\sum_{j=1}^n |L_{d,j}(X)|\} < \infty$, then Assumption (A5$'$) holds with $L_{\hat{\mu}} = \mathbb{E}\{\sum_{j=1}^n |L_{d,j}(X)|\}$. This verification relies only on moment control of the smoothing weights and the intrinsic Lipschitz property of silhouettes. Consequently, for this class of estimators one may dispense with Assumption (A5) and replace Assumption (A5$'$) by the much simpler primitive condition $\mathbb{E}\{\sum_{j=1}^n |L_{d,j}(X)|\} < \infty$. Ordinary least squares is a notable special case.*

The following weak convergence result is the main technical ingredient for our inferential framework. In Appendix C.3 we establish the result under Assumption (A5$'$). Appendix C.4 then shows that the same conclusion holds under Assumption (A5).

**Theorem 5.2.** *Under Assumptions (A1)–(A4) and either (A5′) or (A5), we have*

$$\sqrt{n}\{\widehat{\psi}_{\text{AIPW},d}(t) - \psi_d(t)\} \rightsquigarrow \mathbb{G}_d(t) \quad in \quad \ell^{\infty}(\mathbb{T}),$$

*where $\mathbb{G}_d$ is a mean-zero Gaussian process with $cov\{\mathbb{G}_d(s), \mathbb{G}_d(t)\} = cov\{\varphi_d(s, Z; \eta), \varphi_d(t, Z; \eta)\}$ for all $s, t \in \mathbb{T}$.*

Pointwise $\sqrt{n}$-consistency and asymptotic normality are immediate corollaries of Theorem 5.2. In particular, for each fixed $t \in \mathbb{T}$, $\widehat{\psi}_{\text{AIPW},d}(t)$ attains the semiparametric efficiency bound with asymptotic variance $\text{var}\{\varphi_d(t, Z; \eta)\}$, and achieves $\sqrt{n}$ rates whenever the nuisance errors satisfy a product-rate condition (for example, each converging at $o_{\mathbb{P}}(n^{-1/4})$ in $L_2(\mathbb{P})$). In contrast, the PI and IPW estimators typically incur first-order bias terms driven by $\hat{\mu}_a - \mu_a$ and $\hat{\pi} - \pi$, respectively, and thus generally require substantially stronger nuisance rates to achieve $\sqrt{n}$-consistent inference under flexible nonparametric learning (Kennedy, 2024). In Appendix B, we further provide an explicit bias–variance analysis of the AIPW, IPW, and PI estimators under fixed deviations of $\hat{\mu}_a$ and $\hat{\pi}$ from their true values, demonstrating the doubly robust behavior of the AIPW estimator under nuisance misspecification.

Building on Theorem 5.2, we can construct simultaneous confidence bands over $\mathbb{T}$. There are several approaches to constructing an asymptotically valid $1 - \alpha$ confidence band, depending on the strength of additional assumptions one is willing to impose or the level of computational complexity one is prepared to accommodate. For instance, one may apply the pivotal method proposed by Liebl & Reimherr (2023), or adopt the parametric bootstrap approach developed by Pini & Vantini (2017). For a more detailed discussion and comparison, see Testa et al. (2025).

For many, if not most, practitioners, the primary inferential goal is to determine whether there is any topological effect at all, as captured by persistent homology. Hypothesis tests based on estimated, vectorized topological summaries (e.g., silhouettes, persistence images, or landscapes) generally do not yield valid inference in the metric space of persistence diagrams. In contrast, our framework furnishes a formal test of the null of 'no topological effect'. To this end, we first establish stability bounds for weighted silhouettes, which, to our knowledge, have not previously appeared in the literature.

**Theorem 5.3.** *Let $W_q(\mathcal{D}, \mathcal{D}')$ denote the $q$-Wasserstein distance between two persistence diagrams $\mathcal{D}$ and $\mathcal{D}'$, and let $m^{\star}$ denote the corresponding optimal $W_1$ matching. Assume that*

*(A6) For the power-weighted silhouette defined in equation 2, its corresponding persistence diagram $\mathcal{D}$ is bounded such that for all $p = (a_p, b_p) \in \mathcal{D}$, $-\infty < a_p \leq b_p < \infty$. Moreover, there exists a global constant $L > 0$ such that, for all $p \in \mathcal{D}$ and a given weighting exponent $r$, $\frac{b_p - a_p}{(b_p - a_p)^r} \leq L$. The same holds for $\mathcal{D}'$ as well.*

*Consider weighted silhouette functions $\phi$ and $\phi'$ corresponding to $\mathcal{D}$ and $\mathcal{D}'$. Under Assumption (A6),*

$$\|\phi - \phi'\|_{\infty} \leq (1 + 2Lr\,c^{r-1})\,W_1(\mathcal{D}, \mathcal{D}'),$$

*for some constant $c > 0$ that depends only on $r$ and an upper bound on the persistences in the diagrams $\mathcal{D}, \mathcal{D}'$.*

Appendix C.5 provides the formal definitions of the Wasserstein distance, the associated optimal matching $m^{\star}$, and the constant $c$. Building upon the stability result above and the Gaussian weak convergence of our estimator, our framework provides, to our knowledge, the first formal test of the null hypothesis of no topological effect, with asymptotically correct size and consistency against fixed alternatives, as formally stated below.

**Corollary 5.4.** *Fix $d \in \mathbb{N}_0$ and consider the null hypothesis*

$$H_0: \ W_1(\mathcal{D}_d^1, \mathcal{D}_d^0) = 0 \quad a.s.,$$

*under which $\psi_d(t) = 0$ for all $t \in \mathbb{T}$. Let $T_n := \sqrt{n}\,\|\widehat{\psi}_{\text{AIPW},d}\|_{\infty}$. Under Assumptions (A1)–(A6), we have $T_n \rightsquigarrow \|\mathbb{G}_d\|_{\infty}$, where $\mathbb{G}_d$ is the Gaussian limit in Theorem 5.2. Assume moreover that a Gaussian- or Rademacher-multiplier bootstrap process $\widehat{\mathbb{G}}_{n,d}$ based on the estimated influence function consistently estimates the law of $\|\mathbb{G}_d\|_{\infty}$ conditionally on the data. Let $c_{1-\alpha}$ be the conditional $(1 - \alpha)$-quantile of $\|\widehat{\mathbb{G}}_{n,d}\|_{\infty}$. Then the test that rejects $H_0$ when $T_n > c_{1-\alpha}$ has asymptotic size $\alpha$ and is consistent against any fixed alternative with $\|\psi_d\|_{\infty} > 0$.*

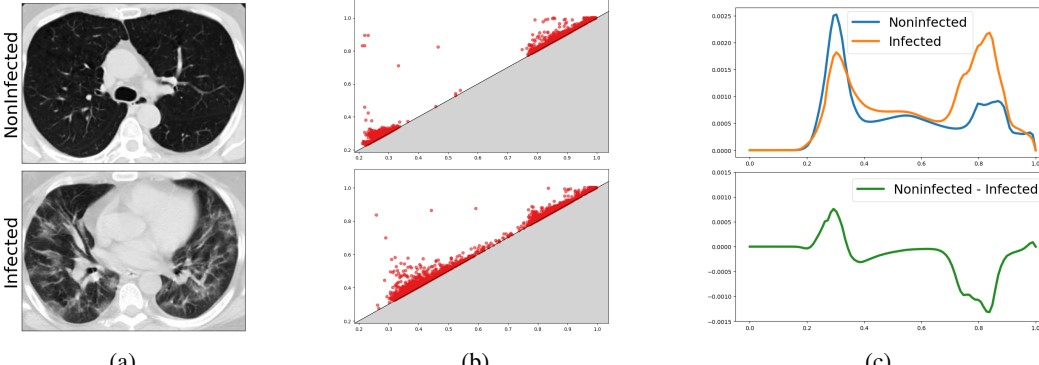

Figure 3: (a) CT-scan images of non-infected (top) and infected (bottom) patients; (b) corresponding 0-dimensional persistence diagrams; (c) average silhouettes of non-infected and infected patients given $r = 0.1$ (top) and difference in average silhouettes between non-infected and infected patients.

Under $H_0$, $T_n = \sqrt{n}\|\widehat{\psi}_{\text{AIPW},d}\|_\infty$ behaves like the supremum norm of the centered empirical-process fluctuation and converges to $\|\mathbb{G}_d\|_\infty$. Under any fixed alternative with $\|\psi_d\|_\infty > 0$, $T_n$ diverges to $+\infty$, so the test rejects with probability tending to one. Because the limit law of $T_n = \sqrt{n}\|\widehat{\psi}_{\text{AIPW},d}\|_\infty$ depends on the unknown distribution of the Gaussian process $\mathbb{G}_d$, we assume a valid multiplier bootstrap so that the conditional $(1 - \alpha)$-quantile of $\|\mathbb{G}_d\|_\infty$ can be consistently approximated from the data, yielding asymptotically correct critical values.

## 6 EXPERIMENTS

To demonstrate the effectiveness of our method, we carry out experiments on two semi-synthetic datasets and one synthetic dataset; results for the synthetic (ORBIT) data are deferred to Appendix D.4 for completeness. For all experiments, we construct a hypothetical dataset $(X, A, Y^0, Y^1)$, where the potential outcome pairs are designed to exhibit distinct topological contrasts across treatment groups. We generate the covariates $X \in \mathbb{R}^5$ from a multivariate Gaussian distribution, imposing a subgroup structure by specifying different mean vectors for each subgroup. Given $X$, treatment $A$ is assigned with probability $\pi(X) = expit(-0.5X_1 - 0.1X_2 + 0.6X_3 + 0.1X_4 + 0.1X_5 + 0.5X_2X_3 - 0.7X_1X_3)$. This treatment mechanism is designed such that one subgroup has a higher probability of receiving treatment than the other. We model the silhouette regression function $\mu_a$ using function-on-scalar regression with a Fourier basis expansion, while the propensity score $\pi$ is estimated via a random forest classifier. Our goal is to estimate the true topological causal effect based on the observable data. All experiments are repeated over 20 simulations. For complete details of the experimental setup, see Appendix D.

**SARS-CoV-2 Dataset.** The first semi-synthetic experiment uses image, possibly in different sizes, with synthesized covariates and treatment assignments while retaining real outcomes, allowing controlled yet realistic evaluation of causal estimators. We employ the SARS-CoV-2 dataset (Soares et al., 2020), which contains CT-scans collected from real patients who are infected or non-infected by COVID-19. Infected patients exhibit high rates of ground-glass opacities and consolidations that appear as isolated regions in CT-scans, which can be captured by 0-dimensional persistence diagrams of Lower-star filtration (Iqbal et al., 2025). In Figure 3a and 3b, the difference between an infected and a non-infected CT scan image is reflected in the associated persistence diagrams. Figure 3c (top) illustrates the average silhouettes of infected and non-infected patients, where the non-infected group exhibits higher values in [0.2, 0.4] and the infected group exhibits higher values in [0.7, 0.9]. Thus, a treatment can be assumed to be more effective when TATE exhibits larger magnitudes over the interval [0.2, 0.4] (positive direction) and [0.7, 0.9] (negative direction), as illustrated in Figure 3c (bottom). In this experiment, the true TATE is known, as we manually construct $(Y^0, Y^1)$ by assigning 500 infected samples to $Y^0$ and subsequently pairing it with $Y^1$, which consist of 75% non-infected and 25% infected samples. We compute the PI, IPW, and AIPW estimators from the observed data and compare their estimates with the true effect.

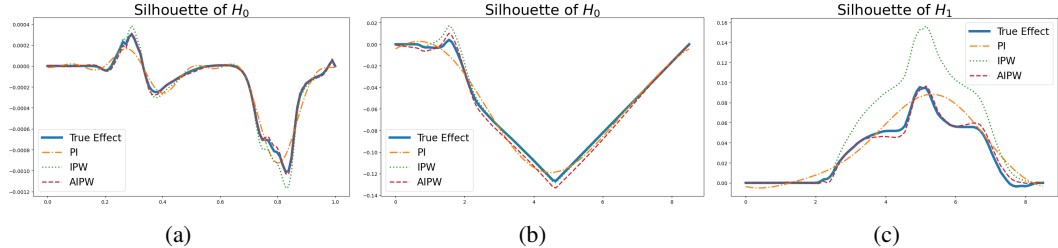

Figure 4: The true silhouette function and its PI, IPW, AIPW estimates. (a) 0-dimensional silhouettes computed from the SARS-CoV-2 dataset, (b) 0-dimensional silhouettes computed from the GEOM-Drugs dataset, and (c) 1-dimensional silhouettes computed from the GEOM-Drugs dataset.

**Results.** The blue curve in Figure 4a shows the true silhouette, which by design clearly reflects a causal effect. Although all three estimators reasonably capture the overall shape of the true target, the IPW estimator systematically overestimates the true treatment effect, whereas the PI estimator underestimates it. In contrast, the AIPW estimator provides an accurate reconstruction of the true silhouette function, achieving minimal bias and closely matching the exact shape of the ground truth.

**GEOM-Drugs Dataset.** Next, we evaluate our framework on graph data through a semi-synthetic experiment with the GEOM-Drugs dataset (Axelrod & Gomez-Bombarelli, 2022), which provides graph-structured representations of molecular compounds. To analyze graph-structured data, we first assign each graph node a scalar weight via a weighted sum of its features, and subsequently define each edge weight as the maximum of the weights of its adjacent nodes; we then apply a sublevel set filtration. Adopting procedures analogous to the previous experiment, we construct $(Y^0, Y^1)$ by taking $Y^0$ to consist of 1000 single-loop graphs and pairing it with $Y^1$, which is a mixture comprising 75% two-loop graphs and 25% single-loop graphs. As before, we compute the PI, IPW, and AIPW estimators from the observed data and compare their estimates to the true effect.

**Results.** All three estimators closely align with the true 0-dimensional silhouette (Figures 4b), which exhibits a negative effect, implying that some connected components were merged under treatment. The true 1-dimensional silhouette (Figure 4c) shows a pronounced positive effect, indicating that treatment induces the formation of new loops, consistent with our experimental design. However, the IPW estimator overestimates the true 1-dimensional treatment effect, while the PI estimator fails to capture the relatively complex curvature. Overall, the AIPW estimator delivers the most accurate and reliable approximation of the true silhouettes, consistent with the theoretical results in Section 5.

## 7 DISCUSSION

We introduce a novel connection between causal inference and TDA, enabling estimation of causal effects that capture not only shifts in mean or variance but also changes in the meaningful intrinsic topological structure of the outcome space. This broadens the scope of causal inference and opens new avenues for investigating causal mechanisms in complex, high-dimensional settings, including those involving unstructured data.

Several caveats and prospective solutions deserve attention, highlighting fruitful directions for future investigation. First, the proposed TATE framework is designed to capture macroscopic topological shifts and may be less informative when the primary interest lies in detecting fine-grained, local changes. In such cases, standard causal estimands could be estimated in parallel if possible, potentially after appropriate preprocessing. Relatedly, as discussed in Section 3, silhouette functions do not exactly quantify the number of changing homological features. Nevertheless, the proposed estimators and the analyses in Sections 4 and 5 extend naturally to individual persistence landscape functions, offering finer topological resolution when required. Lastly, the construction of our estimators can be computationally intensive due to the use of persistent homology. This cost may be mitigated by adopting more efficient topological summaries, such as Euler characteristic curves (Turner et al., 2014). Additional extensions include adapting the framework to more complex causal settings, such as continuous treatments, instrumental variable designs, or longitudinal exposures.

## CODE AVAILABILITY

Python code implementing the proposed methodology is available at `https://github.com/kwangho-joshua-kim/top-causal-effect`.

## ACKNOWLEDGEMENTS

This work was supported by the National Research Foundation of Korea (NRF) grant funded by the Korean government (MSIT)(No. RS-2022-NR068754, RS-2024-00335008, RS-2025-24534596), by a Korea University Grant (K2304801), and by Samsung Science and Technology Foundation under Project Number SSTF-BA2502-01.

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

APPENDIX

## A  STANDARD FILTRATION CONSTRUCTIONS AND RELATED DESCRIPTORS

This appendix provides concrete definitions and examples of the filtration constructions, as well as related topological summaries referenced in Section 2 (Choice of filtration).

**Vietoris–Rips complex.** Let $X$ be a finite set of points equipped with a distance $\mathsf{d}$ (for example, the Euclidean distance when $X \subset \mathbb{R}^m$). For $r > 0$, the Vietoris–Rips complex at scale $r$ is the simplicial complex

$$\mathrm{Rips}(r) = \Big\{ \sigma \subseteq X : \mathsf{d}(u, u') \leq 2r \text{ for all } u, u' \in \sigma \Big\}.$$

If $r_1 \leq r_2$, then $\mathrm{Rips}(r_1) \subseteq \mathrm{Rips}(r_2)$, so $\{\mathrm{Rips}(r) : r \geq 0\}$ forms a natural filtration.

$\alpha$**-complex.** Let $X \subset \mathbb{R}^m$ be a finite point set. For each $u \in X$, let $V_u$ denote the Voronoi cell of $u$,

$$V_u = \big\{ x \in \mathbb{R}^m : \|x - u\| \leq \|x - u'\| \text{ for all } u' \in X \big\},$$

and let $B_u(r) = \{x \in \mathbb{R}^m : \|x - u\| \leq r\}$ be the closed ball of radius $r$ centered at $u$. Define the truncated ball $R_u(r) = B_u(r) \cap V_u$. The $\alpha$-complex at scale $r$ is

$$\mathrm{Alpha}(r) = \Big\{ \sigma \subseteq X : \bigcap_{u \in \sigma} R_u(r) \neq \varnothing \Big\}.$$

As $r$ increases, $\mathrm{Alpha}(r)$ is nested, yielding a filtration $\{\mathrm{Alpha}(r) : r \geq 0\}$.

**Cubical complexes.** A cubical complex is an analogue of a simplicial complex built from axis-aligned cubes, and is well suited for grid-structured data such as digital images. An *elementary interval* is an interval of the form $[l, l]$ or $[l, l + 1]$ for some $l \in \mathbb{Z}$, called *degenerate* and *nondegenerate*, respectively. An *elementary cube* is a finite product $Q = I_1 \times \cdots \times I_n$ of elementary intervals, and its dimension is the number of nondegenerate factors. A *cubical complex* $K$ is a finite collection of elementary cubes such that every face of a cube in $K$ is also in $K$. A filtered cubical complex can be constructed by assigning filtration values to cubes and taking sublevel (or superlevel) subcomplexes.

**Persistence landscapes.** Persistence landscapes (Bubenik et al., 2015) embed persistence diagrams into a functional Hilbert space. Given a persistence diagram $\mathcal{D}$, its persistence landscape is the collection $\{\lambda_k(\cdot; \mathcal{D})\}_{k \in \mathbb{N}}$ defined by

$$\lambda_k(t; \mathcal{D}) = \operatorname*{kmax}_{p \in \mathcal{D}} \Lambda_p(t), \qquad k \in \mathbb{N}, \; t \in \mathbb{T},$$

where $\Lambda_p$ is the tent function in equation 1, kmax denotes the $k$-th largest value (counting multiplicities), and $\mathbb{T} \subset \mathbb{R}$ is a fixed compact interval. When $k$ is fixed, we may write $\lambda(t; \mathcal{D}, k)$ for $\lambda_k(t; \mathcal{D})$, and $\lambda(t; \mathcal{D}_d, k)$ for the landscape associated with the $d$-th diagram. If $\mathcal{D}$ has $N$ off-diagonal points, then $\lambda_k(\cdot; \mathcal{D}) \equiv 0$ for all $k > N$.

## B  BIAS AND VARIANCE ANALYSIS

In this section, we explore how our estimators $\widehat{\psi}_{\mathrm{PI},d}$, $\widehat{\psi}_{\mathrm{IPW},d}$, and $\widehat{\psi}_{\mathrm{AIPW},d}$ perform as the nuisance estimates $\hat{\mu}_a(t, x; d)$ and $\hat{\pi}_a(x)$ deviate from the truth. To this end, we analyze the bias and variance of our estimators in a similar manner to Dudík et al. (2011). Henceforth, we omit the homology dimension $d$ from all notations, as the subsequent results do not depend on $d$. Let $\Delta_a(t, x)$ denote the additive deviation of $\hat{\mu}_a(t, x)$ from $\mu_a(t, x) = \mathbb{E}(\phi(t; \mathcal{D}) \mid X = x, A = a)$, and $\delta_a(x)$ the multiplicative deviation of $\hat{\pi}_a(x)$ from $\pi_a(x) = \mathbb{P}(A = a \mid X = x)$, such that

$$\Delta_a(t, x) = \hat{\mu}_a(t, x) - \mu_a(t, x),$$

$$\delta_a(x) = 1 - \frac{\pi_a(x)}{\hat{\pi}_a(x)}.$$

As discussed in Remark 2, we assume that the estimates $\hat{\mu}_a(t, x)$ and $\hat{\pi}_a(x)$ are fitted from an independent sample via sample splitting, and are thus fixed. For notational convenience, we use the shorthand $\mu_a$ for $\mu_a(t, x)$, $\hat{\mu}_a$ for $\hat{\mu}_a(t, x)$, $\pi_a$ for $\pi_a(x)$, $\hat{\pi}_a$ for $\hat{\pi}_a(x)$, $\mathbb{1}_a$ for $\mathbb{1}(A = a)$, $\phi$ for $\phi(t; \mathcal{D})$, $\Delta_a$ for $\Delta_a(t, x)$, and $\delta_a$ for $\delta_a(x)$.

### B.1 BIAS ANALYSIS

We first analyze the bias of $\widehat{\psi}_{\text{AIPW}}$. Let $\varphi_a := \mu_a + \frac{\mathbb{1}_a}{\pi_a}(\phi - \mu_a)$, and let $\widehat{\varphi}_a$ be its estimate using $\hat{\mu}_a$ and $\widehat{\pi}_a$. Then, the expectation of $\widehat{\varphi}_a$ is

$$\mathbb{E}(\widehat{\varphi}_a) = \mathbb{E}\left\{ \hat{\mu}_a + \frac{\mathbb{1}_a}{\hat{\pi}_a}(\phi - \hat{\mu}_a) \right\}$$

$$= \mathbb{E}\left\{ \mu_a + \Delta_a + \frac{\mathbb{1}_a}{\hat{\pi}_a}(\phi - \mu_a - \Delta_a) \right\}$$

$$= \mathbb{E}(\mu_a) + \mathbb{E}_X\left[ \mathbb{E}\left\{ \Delta_a + \frac{\mathbb{1}_a}{\hat{\pi}_a}(\phi - \mu_a - \Delta_a) \Big| X \right\} \right]$$

$$= \mathbb{E}(\phi^a) + \mathbb{E}_X\left\{ \Delta_a \left( 1 - \frac{\pi_a}{\hat{\pi}_a} \right) + \frac{\pi_a}{\hat{\pi}_a}(\mu_a - \mu_a) \right\}$$

$$= \mathbb{E}(\phi^a) + \mathbb{E}(\Delta_a \delta_a),$$

where the fourth equality follows from the law of total expectation, coupled with the (C1) consistency and (C2) unconfoundedness assumptions from Section 3. From the above formulation, we can derive the bias of $\widehat{\psi}_{\text{AIPW}}$ as follows:

$$\mathbb{E}\left( \widehat{\psi}_{\text{AIPW}} \right) - \psi = \mathbb{E}(\widehat{\varphi}_1 - \widehat{\varphi}_0) - \psi$$

$$= \mathbb{E}(\phi^1 - \phi^0) - \psi + \mathbb{E}(\Delta_1 \delta_1 - \Delta_0 \delta_0)$$

$$= \mathbb{E}(\Delta_1 \delta_1 - \Delta_0 \delta_0).$$

The biases of $\widehat{\psi}_{\text{PI}}$ and $\widehat{\psi}_{\text{IPW}}$ naturally follow from the observation that the PI and IPW estimators are special cases of the AIPW estimator such that $\frac{\mathbb{1}_a}{\hat{\pi}_a}(\phi - \hat{\mu}_a) = 0$ and $\hat{\mu}_a = 0$, respectively.

$$\mathbb{E}\left( \widehat{\psi}_{\text{PI}} \right) - \psi = \mathbb{E}(\Delta_1 - \Delta_0)$$

$$\mathbb{E}\left( \widehat{\psi}_{\text{IPW}} \right) - \psi = -\mathbb{E}(\delta_1 \mu_1 - \delta_0 \mu_0).$$

In general, none of the estimators uniformly dominates the others in terms of bias. However, the bias of the AIPW estimator will be close to 0 if *either* $\hat{\mu}_a \approx \mu_a$ or $\hat{\pi}_a \approx \pi_a$, whereas the PI estimator generally requires $\hat{\mu}_a \approx \mu_a$ and the IPW estimator generally requires $\hat{\pi}_a \approx \pi_a$. Thus, the AIPW estimator can effectively integrate both sources of information for better estimation.

### B.2 VARIANCE ANALYSIS

We now analyze the variance of the estimators. For convenience, we let $\text{Var}(\hat{\psi}) := \text{Cov}(\hat{\psi}(s), \hat{\psi}(t))$ for any $s, t \in \mathbb{T}$, and the subsequent results hold for arbitrary choices of $s$ and $t$. We define $\epsilon_a := \frac{\mathbb{1}_a}{\hat{\pi}_a}(\phi - \mu_a)$ for notational simplicity. Note that from the law of total expectation, $\mathbb{E}(\epsilon_a \mid X) = 0$.

**AIPW.** Given that $\text{Var}(\hat{\psi}_{\text{AIPW}}) = \frac{1}{n}\text{Var}(\hat{\varphi}_1 - \hat{\varphi}_0)$, it suffices to analyze $\text{Var}(\hat{\varphi}_1 - \hat{\varphi}_0)$. Since $\text{Var}(\hat{\varphi}_1 - \hat{\varphi}_0) = \mathbb{E}\{(\hat{\varphi}_1 - \hat{\varphi}_0)^2\} - \mathbb{E}(\hat{\varphi}_1 - \hat{\varphi}_0)^2$ and $\mathbb{E}(\hat{\varphi}_1 - \hat{\varphi}_0)$ has already been derived in the previous section, analyzing $\mathbb{E}\{(\hat{\varphi}_1 - \hat{\varphi}_0)^2\}$ will give us the desired result.

$$\mathbb{E}\{(\hat{\varphi}_1 - \hat{\varphi}_0)^2\} = \mathbb{E}\left[ \left\{ \hat{\mu}_1 + \frac{\mathbb{1}_1}{\hat{\pi}_1}(\phi - \hat{\mu}_1) - \hat{\mu}_0 - \frac{\mathbb{1}_0}{\hat{\pi}_0}(\phi - \hat{\mu}_0) \right\}^2 \right]$$

$$= \mathbb{E}\left[ \left\{ (\mu_1 - \mu_0) + (\epsilon_1 - \epsilon_0) + \Delta_1 \left( 1 - \frac{\mathbb{1}_1}{\hat{\pi}_1} \right) - \Delta_0 \left( 1 - \frac{\mathbb{1}_0}{\hat{\pi}_0} \right) \right\}^2 \right]$$

$$= \mathbb{E}\left\{ (\mu_1 - \mu_0)^2 \right\} + \mathbb{E}\left\{ (\epsilon_1 - \epsilon_0)^2 \right\} + \mathbb{E}\left[ \left\{ \Delta_1 \left( 1 - \frac{\mathbb{1}_1}{\hat{\pi}_1} \right) - \Delta_0 \left( 1 - \frac{\mathbb{1}_0}{\hat{\pi}_0} \right) \right\}^2 \right]$$

$$+ 2\mathbb{E}\left[ (\mu_1 - \mu_0) \left\{ \Delta_1 \left( 1 - \frac{\mathbb{1}_1}{\hat{\pi}_1} \right) - \Delta_0 \left( 1 - \frac{\mathbb{1}_0}{\hat{\pi}_0} \right) \right\} \right],$$

where the expectation of cross products of $\epsilon_1 - \epsilon_0$ is 0 as a consequence of iterated expectation and $\mathbb{E}(\epsilon_a \mid X) = 0$. Next, we further expand the third term and the last term. The third term can be expanded as

$$
\mathbb{E}\left[\left\{\Delta_1\left(1 - \frac{\mathbb{1}_1}{\hat{\pi}_1}\right) - \Delta_0\left(1 - \frac{\mathbb{1}_0}{\hat{\pi}_0}\right)\right\}^2\right]
$$

$$
= \mathbb{E}\left\{\Delta_1^2\left(1 - \frac{\mathbb{1}_1}{\hat{\pi}_1}\right)^2 + \Delta_0^2\left(1 - \frac{\mathbb{1}_0}{\hat{\pi}_0}\right)^2 - 2\Delta_1\Delta_0\left(1 - \frac{\mathbb{1}_1}{\hat{\pi}_1}\right)\left(1 - \frac{\mathbb{1}_0}{\hat{\pi}_0}\right)\right\}
$$

$$
= \mathbb{E}\left\{\Delta_1^2\left(1 - 2\frac{\mathbb{1}_1}{\hat{\pi}_1} + \frac{\mathbb{1}_1}{\hat{\pi}_1^2}\right) + \Delta_0^2\left(1 - 2\frac{\mathbb{1}_0}{\hat{\pi}_0} + \frac{\mathbb{1}_0}{\hat{\pi}_0^2}\right) - 2\Delta_1\Delta_0\left(1 - \frac{\mathbb{1}_1}{\hat{\pi}_1} - \frac{\mathbb{1}_0}{\hat{\pi}_0}\right)\right\}
$$

$$
= \mathbb{E}_X\left\{\Delta_1^2\left(1 - 2\frac{\pi_1}{\hat{\pi}_1} + \frac{\pi_1}{\hat{\pi}_1^2}\right) + \Delta_0^2\left(1 - 2\frac{\pi_0}{\hat{\pi}_0} + \frac{\pi_0}{\hat{\pi}_0^2}\right) - 2\Delta_1\Delta_0\left(1 - \frac{\pi_1}{\hat{\pi}_1} - \frac{\pi_0}{\hat{\pi}_0}\right)\right\}
$$

$$
= \mathbb{E}_X\left\{\Delta_1^2\delta_1^2 + \Delta_1^2\left(\frac{\pi_1(1 - \pi_1)}{\hat{\pi}_1^2}\right) + \Delta_0^2\delta_0^2 + \Delta_0^2\left(\frac{\pi_0(1 - \pi_0)}{\hat{\pi}_0^2}\right) - 2\Delta_1\Delta_0\delta_1\delta_0 + 2\Delta_1\Delta_0\frac{\pi_1\pi_0}{\hat{\pi}_1\hat{\pi}_0}\right\}
$$

$$
= \mathbb{E}_X\left\{(\Delta_1\delta_1 - \Delta_0\delta_0)^2 + \Delta_1^2\left(\frac{\pi_1\pi_0}{\hat{\pi}_1^2}\right) + \Delta_0^2\left(\frac{\pi_1\pi_0}{\hat{\pi}_0^2}\right) + 2\Delta_1\Delta_0\frac{\pi_1\pi_0}{\hat{\pi}_1\hat{\pi}_0}\right\}
$$

$$
= \mathbb{E}_X\left\{(\Delta_1\delta_1 - \Delta_0\delta_0)^2 + \Delta_1^2(1 - \delta_1)^2\left(\frac{\pi_0}{\pi_1}\right) + \Delta_0^2(1 - \delta_0)^2\left(\frac{\pi_1}{\pi_0}\right) + 2\Delta_1\Delta_0(1 - \delta_1)(1 - \delta_0)\right\}
$$

$$
= \mathbb{E}_X\left\{(\Delta_1\delta_1 - \Delta_0\delta_0)^2\right\} + \mathbb{E}_X\left[\left\{\Delta_1(1 - \delta_1)\sqrt{\pi_0/\pi_1} + \Delta_0(1 - \delta_0)\sqrt{\pi_1/\pi_0}\right\}^2\right],
$$

where the third equality follows by iterated expectation. The last term can be expanded as

$$
\mathbb{E}\left[(\mu_1 - \mu_0)\left\{\Delta_1\left(1 - \frac{\mathbb{1}_1}{\hat{\pi}_1}\right) - \Delta_0\left(1 - \frac{\mathbb{1}_0}{\hat{\pi}_0}\right)\right\}\right] = \mathbb{E}_X\left[(\mu_1 - \mu_0)\left\{\Delta_1\left(1 - \frac{\pi_1}{\hat{\pi}_1}\right) - \Delta_0\left(1 - \frac{\pi_0}{\hat{\pi}_0}\right)\right\}\right]
$$

$$
= \mathbb{E}_X\{(\mu_1 - \mu_0)(\Delta_1\delta_1 - \Delta_0\delta_0)\},
$$

where the first equality follows by iterated expectation. Combining the above results, we have

$$
\mathbb{E}\{(\hat{\varphi}_1 - \hat{\varphi}_0)^2\} = \mathbb{E}\left\{(\mu_1 - \mu_0)^2\right\} + 2\mathbb{E}_X\{(\mu_1 - \mu_0)(\Delta_1\delta_1 - \Delta_0\delta_0)\} + \mathbb{E}_X\left\{(\Delta_1\delta_1 - \Delta_0\delta_0)^2\right\}
$$

$$
+ \mathbb{E}\left\{(\epsilon_1 - \epsilon_0)^2\right\} + \mathbb{E}_X\left[\left\{\Delta_1(1 - \delta_1)\sqrt{\pi_0/\pi_1} + \Delta_0(1 - \delta_0)\sqrt{\pi_1/\pi_0}\right\}^2\right]
$$

$$
= \mathbb{E}\left\{(\mu_1 + \Delta_1\delta_1 - \mu_0 - \Delta_0\delta_0)^2\right\} + \mathbb{E}\left\{(\epsilon_1 - \epsilon_0)^2\right\}
$$

$$
+ \mathbb{E}_X\left[\left\{\Delta_1(1 - \delta_1)\sqrt{\pi_0/\pi_1} + \Delta_0(1 - \delta_0)\sqrt{\pi_1/\pi_0}\right\}^2\right].
$$

Having derived $\mathbb{E}\{(\hat{\varphi}_1 - \hat{\varphi}_0)^2\}$, we can now compute the variance of the AIPW estimator. From the bias analysis in the previous section, we know that $\mathbb{E}(\hat{\varphi}_1 - \hat{\varphi}_0) = \mathbb{E}(\mu_1 + \Delta_1\delta_1 - \mu_0 - \Delta_0\delta_0)$. Thus, the variance of the AIPW estimator is

$$
\mathrm{Var}(\hat{\psi}_{\mathrm{AIPW}}) = \frac{1}{n}\mathrm{Var}(\hat{\varphi}_1 - \hat{\varphi}_0)
$$

$$
= \frac{1}{n}\left(\mathbb{E}\{(\hat{\varphi}_1 - \hat{\varphi}_0)^2\} - \mathbb{E}(\hat{\varphi}_1 - \hat{\varphi}_0)^2\right)
$$

$$
= \frac{1}{n}\left(\mathrm{Var}\left(\mu_1 + \Delta_1\delta_1 - \mu_0 - \Delta_0\delta_0\right) + \mathbb{E}\left\{(\epsilon_1 - \epsilon_0)^2\right\}\right.
$$

$$
\left. + \mathbb{E}_X\left[\left\{\Delta_1(1 - \delta_1)\sqrt{\pi_0/\pi_1} + \Delta_0(1 - \delta_0)\sqrt{\pi_1/\pi_0}\right\}^2\right]\right).
$$

**IPW.** To derive the variance of $\hat{\psi}_{\mathrm{IPW}}$, we proceed in an analogous fashion by first analyzing the second moment.

$$
\mathbb{E}\left\{\left(\frac{\mathbb{1}_1}{\hat{\pi}_1}\phi - \frac{\mathbb{1}_0}{\hat{\pi}_0}\phi\right)^2\right\} = \mathbb{E}\left\{\left(\epsilon_1 - \epsilon_0 + \frac{\mathbb{1}_1}{\hat{\pi}_1}\mu_1 - \frac{\mathbb{1}_0}{\hat{\pi}_0}\mu_0\right)^2\right\}
$$

$$
= \mathbb{E}\left[\left\{(\epsilon_1 - \epsilon_0) + (\mu_1 - \mu_0) - \mu_1\left(1 - \frac{\mathbb{1}_1}{\hat{\pi}_1}\right) + \mu_0\left(1 - \frac{\mathbb{1}_0}{\hat{\pi}_0}\right)\right\}^2\right]
$$

$$
= \mathbb{E}\left\{(\epsilon_1 - \epsilon_0)^2\right\} + \mathbb{E}\left\{(\mu_1 - \mu_0)^2\right\} + \mathbb{E}\left[\left\{\mu_1\left(1 - \frac{\mathbb{1}_1}{\hat{\pi}_1}\right) - \mu_0\left(1 - \frac{\mathbb{1}_0}{\hat{\pi}_0}\right)\right\}^2\right]
$$

$$
- 2\mathbb{E}\left[(\mu_1 - \mu_0)\left\{\mu_1\left(1 - \frac{\mathbb{1}_1}{\hat{\pi}_1}\right) - \mu_0\left(1 - \frac{\mathbb{1}_0}{\hat{\pi}_0}\right)\right\}\right],
$$

where the expectation of cross products of $\epsilon_1 - \epsilon_0$ is 0 as a consequence of iterated expectation and $\mathbb{E}(\epsilon_a \mid X) = 0$. Note that the third and the last terms have the same structure as the corresponding terms that appeared in the derivation of the AIPW estimator. Thus, the previous results can be directly applied, and we have

$$
\mathbb{E}\left[\left\{\mu_1\left(1 - \frac{\mathbb{1}_1}{\hat{\pi}_1}\right) - \mu_0\left(1 - \frac{\mathbb{1}_0}{\hat{\pi}_0}\right)\right\}^2\right] = \mathbb{E}_X\left\{(\mu_1\delta_1 - \mu_0\delta_0)^2\right\} + \mathbb{E}_X\left[\left\{\mu_1(1 - \delta_1)\sqrt{\pi_0/\pi_1} + \mu_0(1 - \delta_0)\sqrt{\pi_1/\pi_0}\right\}^2\right],
$$

and

$$
\mathbb{E}\left[(\mu_1 - \mu_0)\left\{\mu_1\left(1 - \frac{\mathbb{1}_1}{\hat{\pi}_1}\right) - \mu_0\left(1 - \frac{\mathbb{1}_0}{\hat{\pi}_0}\right)\right\}\right] = \mathbb{E}_X\{(\mu_1 - \mu_0)(\mu_1\delta_1 - \mu_0\delta_0)\}.
$$

Combining the above results, we obtain

$$
\mathbb{E}\left\{\left(\frac{\mathbb{1}_1}{\hat{\pi}_1}\phi - \frac{\mathbb{1}_0}{\hat{\pi}_0}\phi\right)^2\right\} = \mathbb{E}\left\{(\mu_1 - \mu_0)^2\right\} - 2\mathbb{E}_X\{(\mu_1 - \mu_0)(\mu_1\delta_1 - \mu_0\delta_0)\} + \mathbb{E}_X\left\{(\mu_1\delta_1 - \mu_0\delta_0)^2\right\}
$$

$$
+ \mathbb{E}\left\{(\epsilon_1 - \epsilon_0)^2\right\} + \mathbb{E}_X\left[\left\{\mu_1(1 - \delta_1)\sqrt{\pi_0/\pi_1} + \mu_0(1 - \delta_0)\sqrt{\pi_1/\pi_0}\right\}^2\right]
$$

$$
= \mathbb{E}\left[\left\{(1 - \delta_1)\mu_1 - (1 - \delta_0)\mu_0\right\}^2\right] + \mathbb{E}\left\{(\epsilon_1 - \epsilon_0)^2\right\}
$$

$$
+ \mathbb{E}_X\left[\left\{\mu_1(1 - \delta_1)\sqrt{\pi_0/\pi_1} + \mu_0(1 - \delta_0)\sqrt{\pi_1/\pi_0}\right\}^2\right].
$$

From the bias analysis in the previous section, we know that $\mathbb{E}\left(\frac{\mathbb{1}_1}{\hat{\pi}_1}\phi - \frac{\mathbb{1}_0}{\hat{\pi}_0}\phi\right) = \mathbb{E}\{(1 - \delta_1)\mu_1 - (1 - \delta_0)\mu_0\}$. Thus, the variance of the IPW estimator is

$$
\mathrm{Var}(\hat{\psi}_{\mathrm{IPW}}) = \frac{1}{n}\mathrm{Var}\left(\frac{\mathbb{1}_1}{\hat{\pi}_1}\phi - \frac{\mathbb{1}_0}{\hat{\pi}_0}\phi\right)
$$

$$
= \frac{1}{n}\left(\mathbb{E}\left\{\left(\frac{\mathbb{1}_1}{\hat{\pi}_1}\phi - \frac{\mathbb{1}_0}{\hat{\pi}_0}\phi\right)^2\right\} - \mathbb{E}\left(\frac{\mathbb{1}_1}{\hat{\pi}_1}\phi - \frac{\mathbb{1}_0}{\hat{\pi}_0}\phi\right)^2\right)
$$

$$
= \frac{1}{n}\left(\mathrm{Var}\left\{(1 - \delta_1)\mu_1 - (1 - \delta_0)\mu_0\right\} + \mathbb{E}\left\{(\epsilon_1 - \epsilon_0)^2\right\}\right.
$$

$$
\left. + \mathbb{E}_X\left[\left\{\mu_1(1 - \delta_1)\sqrt{\pi_0/\pi_1} + \mu_0(1 - \delta_0)\sqrt{\pi_1/\pi_0}\right\}^2\right]\right).
$$

We observe that $\mathrm{Var}(\hat{\psi}_{\mathrm{AIPW}})$ and $\mathrm{Var}(\hat{\psi}_{\mathrm{IPW}})$ have similar structures. The first term will generally be of similar magnitude for both estimator when $\hat{\pi} \approx \pi$, but it may be smaller for AIPW if $\hat{\mu} \approx \mu$ is simultaneously satisfied. The second term is identical. In the third term, the only difference is that AIPW involves $\Delta$, whereas IPW replaces it with $\mu$. In general, $\Delta$ will be of smaller magnitude than $\mu$, and IPW is therefore more likely to exhibit larger variance than AIPW.

**PI.** The variance of $\hat{\psi}_{\text{PI}}$ is straightforward:

$$\text{Var}(\hat{\psi}_{\text{PI}}) = \frac{1}{n}\text{Var}(\hat{\mu}_1 - \hat{\mu}_0)$$
$$= \frac{1}{n}\text{Var}(\mu_1 + \Delta_1 - \mu_0 - \Delta_0).$$

Unlike AIPW and IPW estimators, the variance of PI contains only a single term, which corresponds to the first component in the AIPW and IPW variance decompositions. Therefore, PI estimator will generally have smaller variance compared to AIPW and IPW estimators. However, as discussed in the bias analysis, PI estimators are susceptible to high bias if the regression model is misspecified.

## C  PROOFS

**Notation.** We let $\|x\|_q$ denote $L_q$ norm for any fixed vector $x$. For a given function $f$, we use the notation $\|f\|_{\mathbb{P},q} = [\mathbb{P}(|f|^q)]^{1/q} = \left[\int |f(z)|^q d\mathbb{P}(z)\right]^{1/q}$ as the $L_q(\mathbb{P})$-norm of $f$. Also, we let $\mathbb{P}$ denote the conditional expectation given the sample operator $\hat{f}$, as in $\mathbb{P}(\hat{f}) = \int \hat{f}(z)d\mathbb{P}(z)$. Notice that $\mathbb{P}(\hat{f})$ is random only if $\hat{f}$ depends on samples, in which case $\mathbb{P}(\hat{f}) \neq \mathbb{E}(\hat{f})$. Otherwise $\mathbb{P}$ and $\mathbb{E}$ can be used exchangeably. For example, if $\hat{f}$ is constructed on a separate (training) sample $\mathsf{D}^n = (Z_1, ..., Z_n)$, then $\mathbb{P}\left\{\hat{f}(Z)\right\} = \mathbb{E}\left\{\hat{f}(Z) \mid \mathsf{D}^n\right\}$ for a new observation $Z \sim \mathbb{P}$. We let $\mathbb{P}_n$ denote the empirical measure as in $\mathbb{P}_n(f) = \mathbb{P}_n\{(f(Z)\} = \frac{1}{n}\sum_{i=1}^n f(Z_i)$. Lastly, we use the shorthand $a_n \lesssim b_n$ to denote $a_n \leq \mathsf{c}b_n$ for some universal constant $\mathsf{c} > 0$.

### C.1  PROOF OF LEMMA 2.1

*Proof.* For notational convenience, we denote $\phi(t) := \phi(t; \mathcal{D}, r)$ for any fixed persistence diagram $\mathcal{D}$ and the power parameter $r$. Fix $p = (a, b) \in \mathcal{D}$. The tent function $\Lambda_p$ is linear on $[a, (a + b)/2]$ with slope $+1$ and on $[(a + b)/2, b]$ with slope $-1$. Hence, for all $s, t \in \mathbb{R}$,

$$|\Lambda_p(s) - \Lambda_p(t)| \leq |s - t|.$$

Namely, each $\Lambda_p$ is 1–Lipschitz.

Rewrite the weighted silhouette as $\phi(t) = \sum_{j=1}^N \alpha_j \Lambda_{p_j}(t)$ with weights $\alpha_j = w_j/\sum_k w_k \in (0, 1)$ satisfying $\sum_j \alpha_j = 1$. Then it follows that

$$|\phi(s) - \phi(t)| = \left|\sum_j \alpha_j\left(\Lambda_{p_j}(s) - \Lambda_{p_j}(t)\right)\right| \leq \sum_j \alpha_j |s - t| = |s - t|,$$

which shows the silhouette functions is also Lipschitz with constant $L = 1$.

Even when the persistence diagram $\mathcal{D}$ is random, the inequality holds pathwise; thus, taking expectations yields

$$\mathbb{E}\left[\sup_{|s-t|\leq\delta} |\phi(s) - \phi(t)|\right] \leq \delta.$$

$\square$

### C.2  PROOF OF THEOREM 5.1

*Proof.* Because $\pi$ is known, we have

$$\widehat{\psi}_{\text{IPW},d}(t) = \mathbb{P}_n\{\xi_d(Z, t)\}, \qquad \psi_d(t) = \mathbb{P}\{\xi_d(Z, t)\}, \qquad t \in \mathbb{T}.$$

Hence,

$$\sqrt{n}\{\widehat{\psi}_{\text{IPW},d}(t) - \psi_d(t)\} = \sqrt{n}(\mathbb{P}_n - \mathbb{P})\{\xi_d(Z, t)\} = \mathbb{G}_n\,\xi_d(\cdot, t),$$

where $\mathbb{G}_n := \sqrt{n}(\mathbb{P}_n - \mathbb{P})$ denotes the empirical process.

By the given condition that the indexed class $\{\xi_d(\cdot, t) : t \in \mathbb{T}\}$ is $\mathbb{P}$-Donsker (or satisfies any other standard sufficient condition for a functional central limit theorem in $\ell^\infty(\mathbb{T})$), the functional central limit theorem yields

$$\left(\mathbb{G}_n\, \xi_d(\cdot, t)\right)_{t \in \mathbb{T}} \rightsquigarrow \left(\mathbb{G}_d(t)\right)_{t \in \mathbb{T}} \quad \text{in} \quad \ell^\infty(\mathbb{T}),$$

where $\mathbb{G}_d$ is a tight, mean-zero Gaussian process indexed by $\mathbb{T}$.

On the other hand, for any $s, t \in \mathbb{T}$, we have

$$\mathrm{cov}\{\mathbb{G}_d(s), \mathbb{G}_d(t)\} = \mathrm{cov}\big(\xi_d(Z, s), \xi_d(Z, t)\big),$$

which follows from the standard covariance characterization of the Gaussian limit in the Donsker theorem, together with $\sup_{t \in \mathbb{T}} \mathrm{var}\{\xi_d(Z, t)\} < \infty$ (Van Der Vaart & Wellner, 1996, Theorem 2.5.2).

Combining the display above with the empirical-process representation give the desired result:

$$\sqrt{n}\big\{\widehat{\psi}_{\mathrm{IPW},d}(t) - \psi_d(t)\big\} \rightsquigarrow \mathbb{G}_d(t) \quad \text{in} \quad \ell^\infty(\mathbb{T}).$$

$\square$

**Remark 3.** *A sufficient condition for applying a functional central limit theorem to the empirical process indexed by $\{\xi_d(\cdot, t) : t \in \mathbb{T}\}$ is that this class is $\mathbb{P}$-Donsker, which is implied by standard bracketing-entropy bounds under an $L_2(\mathbb{P})$ envelope and suitable measurability. In our setting, these conditions are plausible because $t \mapsto \phi_{i,d}(t)$ is Lipschitz, which transfers to Lipschitz control of $t \mapsto \xi_d(Z, t)$ and thus yields small bracketing entropy, as formalized in the following lemma.*

**Lemma C.1** (A bracketing-entropy sufficient condition for $\{\xi_d(\cdot, t)\}$). *Assume $\mathbb{T} \subset \mathbb{R}$ is compact with diameter $\mathrm{diam}(\mathbb{T}) < \infty$. Then, for all $s, t \in \mathbb{T}$,*

$$|\xi_d(Z, s) - \xi_d(Z, t)| \le \left(\frac{1}{\kappa} + \frac{1}{\kappa}\right)|s - t| = \frac{2}{\kappa}|s - t| \quad a.s.$$

*Moreover, letting $F(Z) := \sup_{t \in \mathbb{T}} |\xi_d(Z, t)|$, we have $F \in L_2(\mathbb{P})$ provided $\sup_{t \in \mathbb{T}} \mathrm{var}\{\xi_d(Z, t)\} < \infty$ and $\sup_{t \in \mathbb{T}} |\mathbb{P}\, \xi_d(Z, t)| < \infty$. In that case, the bracketing numbers satisfy*

$$N_{[]}(\varepsilon\|F\|_{\mathbb{P},2},\, \{\xi_d(\cdot, t) : t \in \mathbb{T}\},\, L_2(\mathbb{P})) \le C\, \varepsilon^{-1}, \qquad 0 < \varepsilon \le 1,$$

*for a constant $C$ depending only on $\mathrm{diam}(\mathbb{T})$ and $\kappa$. Consequently,*

$$\int_0^1 \sqrt{\log N_{[]}(\varepsilon\|F\|_{\mathbb{P},2}, \{\xi_d(\cdot, t)\}, L_2(\mathbb{P}))}\, d\varepsilon \;<\; \infty,$$

*so $\{\xi_d(\cdot, t) : t \in \mathbb{T}\}$ is $\mathbb{P}$-Donsker by a standard bracketing-entropy criterion.*

*Proof.* Fix $s, t \in \mathbb{T}$. By definition,

$$\xi_d(Z, t) = \left\{\frac{A}{\pi(X)} - \frac{1 - A}{1 - \pi(X)}\right\} \phi_{i,d}(t).$$

Under the positivity condition $\kappa \le \pi(X) \le 1 - \kappa$ a.s., the prefactor is bounded in absolute value by $1/\kappa$ when $A = 1$ and by $1/\kappa$ when $A = 0$, hence by $2/\kappa$ uniformly. By virtue of Lemma 2.1, we obtain

$$|\xi_d(Z, s) - \xi_d(Z, t)| \le \left|\frac{A}{\pi(X)} - \frac{1 - A}{1 - \pi(X)}\right| \cdot |\phi_{i,d}(s) - \phi_{i,d}(t)| \le \frac{2}{\kappa}|s - t|.$$

Now cover $\mathbb{T}$ by a grid $\{t_j\}_{j=1}^m$ with mesh size at most $\delta = \varepsilon\|F\|_{\mathbb{P},2}\kappa/2$, so that $m \le 1 + \mathrm{diam}(\mathbb{T})/\delta$. For each $t \in \mathbb{T}$ choose $j(t)$ with $|t - t_{j(t)}| \le \delta$ and define brackets

$$\ell_t(Z) = \xi_d(Z, t_{j(t)}) - \frac{2}{\kappa}\delta, \qquad u_t(Z) = \xi_d(Z, t_{j(t)}) + \frac{2}{\kappa}\delta.$$

Then $\ell_t(Z) \le \xi_d(Z, t) \le u_t(Z)$ a.s. and $\|u_t - \ell_t\|_{\mathbb{P},2} \le (4/\kappa)\delta = 2\varepsilon\|F\|_{\mathbb{P},2}$. Thus $N_{[]}(\varepsilon\|F\|_{\mathbb{P},2}, \cdot, L_2(\mathbb{P})) \le m \lesssim \varepsilon^{-1}$, proving the entropy bound and hence the finite bracketing integral. $\square$

### C.3 PROOF OF THEOREM 5.2 UNDER ASSUMPTION (A5')

Recall that $\varphi$ is the uncentered EIF for the target functional in equation 3), satisfying a Von Mises expansion:

$$\psi(t; \widehat{\mathbb{P}}) - \psi(t; \mathbb{P}) = \int \varphi(t, z; \eta) \, d(\widehat{\mathbb{P}} - \mathbb{P})(z) + R_2(\widehat{\mathbb{P}}, \mathbb{P}), \tag{10}$$

where the second-order remainder term is specified as

$$R_2(\widehat{\mathbb{P}}, \mathbb{P}) = \int \left\{ \frac{1}{\pi(X)} - \frac{1}{\hat{\pi}(X)} \right\} \{\mu_1(t, X; d) - \hat{\mu}_1(t, X; d)\} \pi(X) \, d\mathbb{P}$$
$$- \int \left\{ \frac{1}{1 - \pi(X)} - \frac{1}{1 - \hat{\pi}(X)} \right\} \{\mu_0(t, X; d) - \hat{\mu}_0(t, X; d)\} \{1 - \pi(X)\} \, d\mathbb{P}, \, \forall t \in \mathbb{T}. \tag{11}$$

First, we prove the asymptotic normality of finite-dimensional causal effect projections as stated in the following lemma. The proof follows the argument of Theorem 5.31 in van der Vaart (2002); similar techniques to those have also been employed in Kennedy (2019).

**Lemma C.2** (Asymptotic Normality for Discrete-Time Estimators). *Let $k \in \mathbb{N}$ and $t_1, \dots, t_k \in \mathcal{T}$ be fixed. Throughout in the proof, we write $\widehat{\psi}_d = \widehat{\psi}_{\mathrm{AIPW}, d}$. Define*

$$\widehat{\boldsymbol{\psi}}_{t_1, \dots, t_k} = \left( \widehat{\psi}_d(t_1), \dots, \widehat{\psi}_d(t_k) \right)^\top, \qquad \boldsymbol{\psi}_{t_1, \dots, t_k} = \left( \psi_d(t_1), \dots, \psi_d(t_k) \right)^\top.$$

*Under Assumptions (A1)-(A5), we have*

$$\sqrt{n} \left( \widehat{\boldsymbol{\psi}}_{t_1, \dots, t_k} - \boldsymbol{\psi}_{t_1, \dots, t_k} \right) \xrightarrow{d} N(0, \Sigma_{t_1, \dots, t_k}),$$

*where the covariance matrix is given by*

$$\Sigma_{t_1, \dots, t_k} = \mathbb{E} \left[ \boldsymbol{\varphi}_{t_1, \dots, t_k}(\mathcal{D}) \boldsymbol{\varphi}_{t_1, \dots, t_k}(\mathcal{D})^\top \right],$$

*where we write $\boldsymbol{\varphi}_{t_1, \dots, t_k}(\mathcal{D}) = (\varphi(t_1; \mathcal{D}), \dots, \varphi(t_k; \mathcal{D}))^\top$.*

*Proof.* For notational simplicity, let $\boldsymbol{\varphi} = \boldsymbol{\varphi}_{t_1, \dots, t_k}(\mathcal{D})$ and $\widehat{\boldsymbol{\varphi}} = \widehat{\boldsymbol{\varphi}}_{t_1, \dots, t_k}(\mathcal{D}) = (\widehat{\varphi}(t_1; \mathcal{D}), \dots, \widehat{\varphi}(t_k; \mathcal{D}))^\top$. Recall that $\widehat{\varphi}(t; \mathcal{D})$ is defined by

$$\widehat{\varphi}(t; \mathcal{D}) = \hat{\mu}_1(t, X; d) - \hat{\mu}_0(t, X; d) + \left\{ \frac{A}{\hat{\pi}(X)} - \frac{1 - A}{1 - \hat{\pi}(X)} \right\} \{\phi(t; \mathcal{D}_d) - \hat{\mu}_A(t, X; d)\},$$

where all the nuisance estimators are constructed on a separate, independent sample (Assumption (A2)). Then, consider the following decomposition:

$$\sqrt{n} \left( \widehat{\boldsymbol{\psi}}_{t_1, \dots, t_k} - \boldsymbol{\psi}_{t_1, \dots, t_k} \right) = \sqrt{n}(\mathbb{P}_n - \mathbb{P})\boldsymbol{\varphi} + \sqrt{n}(\mathbb{P}_n - \mathbb{P})(\widehat{\boldsymbol{\varphi}} - \boldsymbol{\varphi}) + \mathbb{P}(\widehat{\boldsymbol{\varphi}} - \boldsymbol{\varphi}). \tag{12}$$

By the multivariate central limit theorem, the first term in equation 12 converges to a multivariate Normal distribution with mean 0 and covariance matrix equal to $\Sigma_{t_1, \dots, t_k}$. The third term in equation 12 is simply $\sqrt{n}R_2$ where $R_2$ is the remainder term specified in equation 11. Under the rate condition in Assumption (A4), it follows that $\sqrt{n}R_2 = o_{\mathbb{P}}(1)$. Thus, it suffices to show that the second empirical process term is negligible, i.e., of order $o_{\mathbb{P}}(1)$. By the consistency condition (A3) and the triangle inequality,

$$\|\widehat{\boldsymbol{\varphi}} - \boldsymbol{\varphi}\|_{\mathbb{P}, 2} = O \left( \sum_{j=1}^k \|\widehat{\varphi}(t_j) - \varphi(t_j)\|_{\mathbb{P}, 2} \right) = o_{\mathbb{P}}(1).$$

Applying the multidimensional Chebyshev's inequality to the proof of Lemma 2 in Kennedy et al. (2020), it is immediate to get

$$(\mathbb{P}_n - \mathbb{P})(\widehat{\boldsymbol{\varphi}} - \boldsymbol{\varphi}) = O_{\mathbb{P}} \left( \frac{\|\widehat{\boldsymbol{\varphi}} - \boldsymbol{\varphi}\|_{\mathbb{P}, 2}}{\sqrt{n}} \right) = o_{\mathbb{P}}(n^{-1/2}).$$

Putting all the pieces together into equation 12, the result follows by Slutsky's theorem. $\qquad \square$

We are now in a position to prove Theorem 5.2. Here, we impose Assumption (A5') and follow the proof strategy of Testa et al. (2025). We then show that the same result holds under the weaker conditions from Assumption (A5), in Section C.4.

*Proof.* First, we show the stochastic equicontinuity of $\widehat{\psi}$:

$$\lim_{\delta \to 0} \limsup_{n \to \infty} \mathbb{P}\left[\sup_{|s-t| \leq \delta} \left|\widehat{\psi}(s) - \widehat{\psi}(t)\right| \geq \varepsilon\right] = 0,$$

for every $\varepsilon > 0$. Fix $\delta > 0$ and write

$$w(\widehat{\psi}; \delta) = \sup_{|s-t| \leq \delta} \left|\widehat{\psi}(s) - \widehat{\psi}(t)\right|.$$

For $a \in \{0, 1\}$ set $\Delta_a(s, t) = \widehat{\mu}_a(s, X; d) - \widehat{\mu}_a(t, X; d)$ and $R(s, t) = \phi(s; \mathcal{D}_d) - \phi(t; \mathcal{D}_d) - \left\{\widehat{\mu}_A(s, X; d) - \widehat{\mu}_A(t, X; d)\right\}$. Then

$$\widehat{\psi}(s) - \widehat{\psi}(t) = \Delta_1(s, t) - \Delta_0(s, t) + \left\{\frac{A}{\widehat{\pi}(X)} - \frac{1-A}{1-\widehat{\pi}(X)}\right\} R(s, t).$$

By Assumption (A5), $\mathbb{P}\left[\sup_{|s-t| \leq \delta} |\Delta_a(s, t)|\right] \leq L_{\widehat{\mu}}\delta, \forall a \in \mathcal{A}$. Since persistent pairs with infinite lifespan are excluded from consideration, we have $\|\phi\|_\infty < \infty$. Further, by Lemma 2.1, $\phi(t)$ is 1-Lipschitz continuous in $t$, yielding $|\phi(s; \mathcal{D}_d) - \phi(t; \mathcal{D}_d)| \leq |s - t|$. Thus, it follows that $\mathbb{P}\left[\sup_{|s-t| \leq \delta} |R(s, t)|\right] \leq (L_{\widehat{\mu}} + 1)\delta$. Therefore, by the Markov inequality, for any fixed $\varepsilon > 0$, we have

$$\mathbb{P}\left[w(\widehat{\psi}; \delta) \geq \varepsilon\right] \leq \frac{\mathbb{P}\left[w(\widehat{\psi}; \delta)\right]}{\varepsilon} \lesssim \frac{(3L_{\widehat{\mu}} + 1)\delta}{\varepsilon},$$

where the second inequality follows by the triangle inequality and Assumption (A1). Finally, taking $\delta \to 0$ and then $\limsup_{n \to \infty}$ yields $\lim_{\delta \to 0} \limsup_{n \to \infty} \mathbb{P}\left[w(\widehat{\psi}; \delta) \geq \varepsilon\right] = 0$ for every $\varepsilon > 0$, as desired. Having established the stochastic equicontinuity of $\widehat{\psi}$, the result follows directly from Lemma C.2 and Theorem 7.5 of Billingsley (1999). $\square$

## C.4 PROOF OF THEOREM 5.2 UNDER ASSUMPTION (A5)

*Proof.* The finite-dimensional convergence argument in Section C.3 is unchanged, so it suffices to establish stochastic equicontinuity.

Fix $\delta > 0$ and write

$$\sup_{|s-t| \leq \delta} \left|\widehat{\psi}(s) - \widehat{\psi}(t)\right| \leq \mathbb{P}_n\left[\sup_{|s-t| \leq \delta} |\widehat{\varphi}_d(s, Z) - \widehat{\varphi}_d(t, Z)|\right].$$

Decompose $\widehat{\varphi}_d(t, Z) = \varphi_d(t, Z; \eta) + \{\widehat{\varphi}_d(t, Z) - \varphi_d(t, Z; \eta)\}$ to obtain

$$\sup_{|s-t| \leq \delta} |\widehat{\varphi}_d(s, Z) - \widehat{\varphi}_d(t, Z)| \leq \sup_{|s-t| \leq \delta} |\varphi_d(s, Z; \eta) - \varphi_d(t, Z; \eta)| + 2\sup_{u \in \mathbb{T}} |\widehat{\varphi}_d(u, Z) - \varphi_d(u, Z; \eta)|.$$

For the first term, Assumption (A5) implies that $t \mapsto \mu_a(t, x; d)$ is Lipschitz uniformly in $x$, and Lemma 2.1 gives Lipschitz continuity of $t \mapsto \phi_{i,d}(t)$. Together with the boundedness of $1/\pi(X)$ and $1/\{1 - \pi(X)\}$ implied by positivity, this yields a constant $L_\varphi < \infty$ such that

$$\sup_{|s-t| \leq \delta} |\varphi_d(s, Z; \eta) - \varphi_d(t, Z; \eta)| \leq L_\varphi \delta \quad \text{a.s.}$$

For the second term, uniform consistency in Assumption (A5) and bounded denominators (Assumption (A1)) imply

$$\Delta_n := \sup_{u \in \mathbb{T}} |\widehat{\varphi}_d(u, Z) - \varphi_d(u, Z; \eta)| = o_{\mathbb{P}}(1).$$

Therefore,

$$\sup_{|s-t| \leq \delta} |\widehat{\varphi}_d(s, Z) - \widehat{\varphi}_d(t, Z)| \leq L_\varphi \delta + 2\Delta_n.$$

Choose $\delta = \delta_n \to 0$ such that $L_\varphi \delta_n \leq \varepsilon/2$. Then

$$\mathbb{P}\left(\sup_{|s-t|\leq\delta_n}\left|\widehat{\psi}(s) - \widehat{\psi}(t)\right| > \varepsilon\right) \leq \mathbb{P}(2\Delta_n > \varepsilon/2) \to 0,$$

which establishes stochastic equicontinuity. The remaining tightness and conclusion follow as in Section C.3, completing the proof under Assumption (A5). $\square$

The finite-dimensional convergence argument in Section C.3 is unchanged, so it suffices to establish stochastic equicontinuity of the process $t \mapsto \sqrt{n}\{\widehat{\psi}(t) - \psi_d(t)\}$ in $\ell^\infty(\mathbb{T})$.

Throughout, recall that the proposed estimator is

$$\widehat{\psi}(t) = \mathbb{P}_n\{\widehat{\varphi}_d(t, Z)\}, \qquad \widehat{\varphi}_d(t, Z) \equiv \varphi_d(t, Z; \widehat{\eta}),$$

where $\widehat{\eta} = (\widehat{\pi}, \widehat{\mu}_0, \widehat{\mu}_1)$ is fit on an independent sample from $\widehat{\mathbb{P}}$, where $\mathbb{P}_n \perp\!\!\!\perp \widehat{\mathbb{P}}$ by Assumption (A2). We also write $\varphi_d(t, Z) \equiv \varphi_d(t, Z; \eta)$ for the true EIF with $\eta = (\pi, \mu_0, \mu_1)$.

Fix $\delta > 0$. Since $\widehat{\psi}(t) = \mathbb{P}_n\{\widehat{\varphi}_d(t, Z)\}$, we have

$$\sup_{|s-t|\leq\delta}\left|\widehat{\psi}(s) - \widehat{\psi}(t)\right| = \sup_{|s-t|\leq\delta}\left|\mathbb{P}_n\{\widehat{\varphi}_d(s, Z) - \widehat{\varphi}_d(t, Z)\}\right|$$
$$\leq \sup_{|s-t|\leq\delta}\mathbb{P}_n\left|\widehat{\varphi}_d(s, Z) - \widehat{\varphi}_d(t, Z)\right| \leq \mathbb{P}_n\left[\sup_{|s-t|\leq\delta}\left|\widehat{\varphi}_d(s, Z) - \widehat{\varphi}_d(t, Z)\right|\right]. \tag{13}$$

Next, for any $t \in \mathbb{T}$,
$$\widehat{\varphi}_d(t, Z) = \varphi_d(t, Z) + \{\widehat{\varphi}_d(t, Z) - \varphi_d(t, Z)\},$$

and thereby, $\forall s, t \in \mathbb{T}$,

$$\left|\widehat{\varphi}_d(s, Z) - \widehat{\varphi}_d(t, Z)\right| \leq \left|\varphi_d(s, Z) - \varphi_d(t, Z)\right| + \left|\widehat{\varphi}_d(s, Z) - \varphi_d(s, Z)\right| + \left|\widehat{\varphi}_d(t, Z) - \varphi_d(t, Z)\right|$$
$$\leq \left|\varphi_d(s, Z) - \varphi_d(t, Z)\right| + 2\sup_{u\in\mathbb{T}}\left|\widehat{\varphi}_d(u, Z) - \varphi_d(u, Z)\right|. \tag{14}$$

Recall that

$$\varphi_d(t, Z) = \mu_1(t, X; d) - \mu_0(t, X; d) + \left\{\frac{A}{\pi(X)} - \frac{1-A}{1-\pi(X)}\right\}\{\phi_{i,d}(t) - \mu_A(t, X; d)\}.$$

By Assumption (A5), for each $a \in \{0, 1\}$, $t \mapsto \mu_a(t, x; d)$ is Lipschitz on $\mathbb{T}$ uniformly over $x$, so there exists $L_\mu < \infty$ such that

$$\sup_{x\in\mathcal{X}}\left|\mu_a(s, x; d) - \mu_a(t, x; d)\right| \leq L_\mu|s - t|.$$

By Lemma 2.1, $t \mapsto \phi_{i,d}(t)$ is 1-Lipschitz, so $|\phi_{i,d}(s) - \phi_{i,d}(t)| \leq |s - t|$. Moreover, the positivity condition stated elsewhere implies that $\|1/\pi\|_\infty < \infty$ and $\|1/(1-\pi)\|_\infty < \infty$; set

$$C_\pi := \left\|\frac{1}{\pi}\right\|_\infty + \left\|\frac{1}{1-\pi}\right\|_\infty < \infty.$$

Then for any $s, t \in \mathbb{T}$,

$$|\varphi_d(s, Z) - \varphi_d(t, Z)| \leq |\mu_1(s, X; d) - \mu_1(t, X; d)| + |\mu_0(s, X; d) - \mu_0(t, X; d)|$$
$$+ \left|\frac{A}{\pi(X)} - \frac{1-A}{1-\pi(X)}\right|\left(|\phi_{i,d}(s) - \phi_{i,d}(t)| + |\mu_A(s, X; d) - \mu_A(t, X; d)|\right)$$
$$\leq 2L_\mu|s - t| + C_\pi\{|s - t| + L_\mu|s - t|\} = L_\varphi|s - t|,$$

where $L_\varphi := 2L_\mu + C_\pi(1 + L_\mu) < \infty$. Consequently,

$$\sup_{|s-t|\leq\delta}|\varphi_d(s, Z) - \varphi_d(t, Z)| \leq L_\varphi\delta \qquad \text{a.s.} \tag{15}$$

From equation 14, define

$$\Delta_n(Z) := \sup_{u \in \mathbb{T}} |\widehat{\varphi}_d(u, Z) - \varphi_d(u, Z)|.$$

Note that

$$\begin{aligned}
\widehat{\varphi}_d(t, Z) - \varphi_d(t, Z) &= \{\hat{\mu}_1(t, X; d) - \mu_1(t, X; d)\} - \{\hat{\mu}_0(t, X; d) - \mu_0(t, X; d)\} \\
&\quad + \left(\frac{A}{\hat{\pi}(X)} - \frac{A}{\pi(X)}\right)\{\phi_{i,d}(t) - \hat{\mu}_1(t, X; d)\} \\
&\quad - \left(\frac{1-A}{1-\hat{\pi}(X)} - \frac{1-A}{1-\pi(X)}\right)\{\phi_{i,d}(t) - \hat{\mu}_0(t, X; d)\} \\
&\quad + \left\{\frac{A}{\pi(X)} - \frac{1-A}{1-\pi(X)}\right\}\{\mu_A(t, X; d) - \hat{\mu}_A(t, X; d)\}.
\end{aligned}$$

Now, take $\sup_{t \in \mathbb{T}} |\cdot|$ and use the triangle inequality. Assumption (A1) gives bounded denominators for $\hat{\pi}$, and the positivity condition gives bounded denominators for $\pi$. Also, we have $\|\phi_{i,d}\|_\infty < \infty$. Therefore there exists an almost surely finite random constant $C(Z)$ (depending only on $\|\phi_{i,d}\|_\infty$, $\|1/\pi\|_\infty$, $\|1/(1-\pi)\|_\infty$, $\|1/\hat{\pi}\|_\infty$, and $\|1/(1-\hat{\pi})\|_\infty$) such that

$$\Delta_n(Z) = \sup_{t \in \mathbb{T}} |\widehat{\varphi}_d(t, Z) - \varphi_d(t, Z)| \le C(Z)\left\{\max_{a \in \{0,1\}} \sup_{t \in \mathbb{T}} |\hat{\mu}_a(t, X; d) - \mu_a(t, X; d)| + |\hat{\pi}(X) - \pi(X)|\right\}. \tag{16}$$

Now Assumption (A5) implies the uniform-in-$(t, x)$ consistency $\sup_{t,x} |\hat{\mu}_a(t, x; d) - \mu_a(t, x; d)| = o_\mathbb{P}(1)$ and $\sup_x |\hat{\pi}(x) - \pi(x)| = o_\mathbb{P}(1)$, hence the right-hand side of equation 16 is $o_\mathbb{P}(1)$. Therefore,

$$\Delta_n(Z) = o_\mathbb{P}(1). \tag{17}$$

Combine equation 14, equation 15, and equation 17 to obtain

$$\sup_{|s-t| \le \delta} |\widehat{\varphi}_d(s, Z) - \widehat{\varphi}_d(t, Z)| \le L_\varphi \delta + 2\Delta_n(Z).$$

Plugging into equation 13 yields

$$\sup_{|s-t| \le \delta} |\widehat{\psi}(s) - \widehat{\psi}(t)| \le L_\varphi \delta + 2\mathbb{P}_n\{\Delta_n(Z)\}.$$

Since $\mathbb{P}_n\{\Delta_n(Z)\} \to 0$ in probability by equation 17, for any $\varepsilon > 0$ choose a deterministic sequence $\delta_n \to 0$ such that $L_\varphi \delta_n \le \varepsilon/2$. Then

$$\mathbb{P}\left(\sup_{|s-t| \le \delta_n} |\widehat{\psi}(s) - \widehat{\psi}(t)| > \varepsilon\right) \le \mathbb{P}(2\mathbb{P}_n\{\Delta_n(Z)\} > \varepsilon/2) \to 0,$$

which verifies stochastic equicontinuity, thus completing the proof of Theorem 5.2 under Assumption (A5).

## C.5 PROOF OF THEOREM 5.3

We first introduce several definitions that will be used in the development of our stability results.

**Definition C.1** (Matching). *Let $\triangle$ denote the diagonal of a persistence diagram. A* matching *between two persistence diagrams $\mathcal{D}$ and $\mathcal{D}'$ is defined as a subset $m \subset \mathcal{D} \times \mathcal{D}'$ such that every point in $\mathcal{D} \setminus \triangle$ and $\mathcal{D}' \setminus \triangle$ appears exactly once in $m$.*

**Definition C.2** (Wasserstein distance). *The $q$-th Wasserstein distance* between two persistence diagrams $\mathcal{D}$ and $\mathcal{D}'$ is

$$W_q(\mathcal{D}, \mathcal{D}') = \inf_{\text{matching } m} \left(\sum_{(p,p') \in m} \|p - p'\|_\infty^q\right)^{1/q}.$$

We denote the matching $m$ that satisfies Definition C.2 as the *optimal $W_q$ matching.*

Let $\phi = \phi(t; \mathcal{D}, r)$ and $\phi' = \phi(t; \mathcal{D}', r)$ be the corresponding power-weighted silhouettes for persistence diagrams $\mathcal{D}$ and $\mathcal{D}'$, respectively. Given $p = (a_p, b_p) \in \mathcal{D}$ and $p' = (a_{p'}, b_{p'}) \in \mathcal{D}'$, we denote the power weights corresponding to $p$ and $p'$ as $w_p = |b_p - a_p|^r$ and $w_{p'} = |b_{p'} - a_{p'}|^r$, respectively, where $r > 0$. Note that under the diagram boundedness condition of Assumption (A6), the absolute value in the power weights may be omitted. Hence, we assume $w_p = (b_p - a_p)^r$ and $w_{p'} = (b_{p'} - a_{p'})^r$ throughout this section.

We now establish the following lemma, which serves as a preliminary result essential for the development of the main theorem.

**Lemma C.3.** *Given a matching $m \subset \mathcal{D} \times \mathcal{D}'$, for any $(p, p') \in m$,*
$$|w_p - w_{p'}| \leq 2rc_{pp'}^{r-1} \cdot \|p - p'\|_\infty,$$
*where $c_{pp'} \in (\min\{b_p - a_p,\ b_{p'} - a_{p'}\}, \max\{b_p - a_p,\ b_{p'} - a_{p'}\})$ is some positive constant that satisfies the Mean Value Theorem for the function $g(x) = x^r, x \in [0, \infty), r > 0$, given points $p$ and $p'$. Consequently,*
$$|w_p - w_{p'}| \leq 2rc^{r-1} \cdot \|p - p'\|_\infty,$$
*for*
$$c = \max_{(p,p') \in m} \max\{b_p - a_p,\ b_{p'} - a_{p'}\} = \max\{b_x - a_x : x \in \mathcal{D} \cup \mathcal{D}'\},$$
*so $c$ depends only on the weighting exponent $r$ (through the bound) and on the largest lifetime across $\mathcal{D}$ and $\mathcal{D}'$.*

*Proof.*
$$
\begin{aligned}
|w_p - w_{p'}| &= |(b_p - a_p)^r - (b_{p'} - a_{p'})^r| \\
&= |r\, c_{pp'}^{r-1}\{(b_p - a_p) - (b_{p'} - a_{p'})\}| \\
&= r\, c_{pp'}^{r-1}|(b_p - a_p) - (b_{p'} - a_{p'})| \\
&\leq r\, c_{pp'}^{r-1}\{|a_p - a_{p'}| + |b_p - b_{p'}|\} \\
&\leq 2r\, c_{pp'}^{r-1}\max\{|a_p - a_{p'}|, |b_p - b_{p'}|\} \\
&= 2r\, c_{pp'}^{r-1}\|p - p'\|_\infty,
\end{aligned}
$$
where the second equality uses the Mean Value Theorem. $\qquad\square$

Notice that when $r = 1$ the bound does not involve $c$ (cf. the special case in the theorem).

Now we give the proof of Theorem 5.3.

*Proof.* Let $w_p = (b_p - a_p)^r$, $w_{p'} = (b_{p'} - a_{p'})^r$, and define
$$S = \sum_{p \in \mathcal{D}} w_p, \qquad S' = \sum_{p' \in \mathcal{D}'} w_{p'}, \qquad \phi = \frac{1}{S}\sum_{p \in \mathcal{D}} w_p \Lambda_p, \qquad \phi' = \frac{1}{S'}\sum_{p' \in \mathcal{D}'} w_{p'} \Lambda_{p'},$$
where $\Lambda_p$ denotes the tent function centered at $p$ such that $\|\Lambda_p\|_\infty = (b_p - a_p)/2$. Augmenting the diagrams with the diagonal if needed, the optimal $W_1$ matching $m^*$ is a bijection between $\mathcal{D}$ and $\mathcal{D}'$, hence
$$\|\phi - \phi'\|_\infty = \left\|\sum_{(p,p')\in m^*}\left(\frac{w_p\Lambda_p}{S} - \frac{w_{p'}\Lambda_{p'}}{S'}\right)\right\|_\infty \leq \sum_{(p,p')\in m^*}\left\|\frac{w_p\Lambda_p}{S} - \frac{w_{p'}\Lambda_{p'}}{S'}\right\|_\infty.$$
Split each summand as
$$\left\|\frac{w_p\Lambda_p}{S} - \frac{w_{p'}\Lambda_{p'}}{S'}\right\|_\infty \leq \left\|\frac{w_p}{S}(\Lambda_p - \Lambda_{p'})\right\|_\infty + \|\Lambda_{p'}\|_\infty\left|\frac{w_p}{S} - \frac{w_{p'}}{S'}\right|.$$
Summing over $(p, p') \in m^*$ yields
$$\|\phi - \phi'\|_\infty \leq T_1 + T_2, \qquad T_1 := \frac{1}{S}\sum_{(p,p')\in m^*}w_p\|\Lambda_p - \Lambda_{p'}\|_\infty, \qquad T_2 := \sum_{(p,p')\in m^*}\|\Lambda_{p'}\|_\infty\left|\frac{w_p}{S} - \frac{w_{p'}}{S'}\right|.$$

**Bound for $T_1$.** By the 1-Lipschitz property of the tent functions, we have $\|\Lambda_p - \Lambda_{p'}\|_\infty \le \|p - p'\|_\infty$. Therefore,

$$T_1 \le \frac{1}{S} \sum_{(p,p') \in m^*} w_p \|p - p'\|_\infty \le \frac{1}{S} \Big( \sum_{p \in \mathcal{D}} w_p \Big) \Big( \sum_{(p,p') \in m^*} \|p - p'\|_\infty \Big) = W_1(\mathcal{D}, \mathcal{D}'),$$

where the second inequality follows by a simple corollary of Hölder's inequality:

$$\sum_i \beta_i x_i \le \|\beta\|_1 \|x\|_\infty \le \|\beta\|_1 \|x\|_1 = \Big( \sum_i \beta_i \Big) \Big( \sum_i x_i \Big),$$

for $\beta_i, x_i \ge 0$, and the last equality is the definition of the $W_1$ cost of $m^*$.

**Bound for $T_2$.** Observe the algebraic identity

$$\Big| \frac{w_p}{S} - \frac{w_{p'}}{S'} \Big| = \frac{|(S' - S)w_p + S(w_p - w_{p'})|}{SS'} \le \frac{|S' - S| \, w_p}{SS'} + \frac{|w_p - w_{p'}|}{S'}.$$

Hence $T_2 \le T_{2a} + T_{2b}$, where

$$T_{2a} := \sum_{(p,p') \in m^*} \|\Lambda_{p'}\|_\infty \frac{|S' - S| \, w_p}{SS'}, \qquad T_{2b} := \sum_{(p,p') \in m^*} \|\Lambda_{p'}\|_\infty \frac{|w_p - w_{p'}|}{S'}.$$

We first tackle the term $T_{2a}$. Using Hölder's inequality to separate the sums and cancel $S$, we have

$$T_{2a} \le \Big( \sum_{p' \in \mathcal{D}'} \|\Lambda_{p'}\|_\infty \Big) \frac{|S' - S|}{S'}.$$

Since $S' - S = \sum_{(p,p') \in m^*} (w_{p'} - w_p)$, the triangle inequality and Lemma C.3 yield

$$|S' - S| \le \sum_{(p,p') \in m^*} |w_p - w_{p'}| \le \sum_{(p,p') \in m^*} 2r \, c_{pp'}^{r-1} \|p - p'\|_\infty \le 2r \, c^{r-1} \sum_{(p,p') \in m^*} \|p - p'\|_\infty.$$

Moreover, $\sum_{p' \in \mathcal{D}'} \|\Lambda_{p'}\|_\infty = \frac{1}{2} \sum_{p' \in \mathcal{D}'} (b_{p'} - a_{p'})$ and $S' = \sum_{p' \in \mathcal{D}'} (b_{p'} - a_{p'})^r$, so by the boundedness condition from Assumption (A6),

$$\frac{\sum_{p' \in \mathcal{D}'} \|\Lambda_{p'}\|_\infty}{S'} = \frac{1}{2} \cdot \frac{\sum_{p'} (b_{p'} - a_{p'})}{\sum_{p'} (b_{p'} - a_{p'})^r} \le \frac{L}{2}.$$

Therefore

$$T_{2a} \le \frac{L}{2} 2r \, c^{r-1} \sum_{(p,p') \in m^*} \|p - p'\|_\infty = L \, r \, c^{r-1} \, W_1(\mathcal{D}, \mathcal{D}').$$

The analysis of $T_{2b}$ follows the same steps as above. Again by Lemma C.3 and the ratio boundedness condition from Assumption (A6),

$$T_{2b} \le \frac{\sum_{p' \in \mathcal{D}'} \|\Lambda_{p'}\|_\infty}{S'} \sum_{(p,p') \in m^*} |w_p - w'_{p'}| \le \frac{L}{2} \sum_{(p,p') \in m^*} 2r \, c_{pp'}^{r-1} \|p - p'\|_\infty \le L \, r \, c^{r-1} \, W_1(\mathcal{D}, \mathcal{D}').$$

Combining the bounds for $T_{2a}$ and $T_{2b}$ gives

$$T_2 \le 2L \, r \, c^{r-1} \, W_1(\mathcal{D}, \mathcal{D}').$$

Putting together the bounds for $T_1$ and $T_2$, we obtain

$$\|\phi - \phi'\|_\infty \le \big( 1 + 2L \, r \, c^{r-1} \big) W_1(\mathcal{D}, \mathcal{D}').$$

**Special case $r = 1$.** When $r = 1$, Lemma C.3 gives $|w_p - w_{p'}| \le 2\|p - p'\|_\infty$ and

$$\frac{\sum_{p' \in \mathcal{D}'} \|\Lambda_{p'}\|_\infty}{S'} = \frac{\sum_{p'} (b_{p'} - a_{p'})/2}{\sum_{p'} (b_{p'} - q_{p'})} = \frac{1}{2},$$

so each of $T_{2a}$ and $T_{2b}$ is bounded by $W_1(\mathcal{D}, \mathcal{D}')$, while $T_1 \le W_1(\mathcal{D}, \mathcal{D}')$, which yields

$$\|\phi - \phi'\|_\infty \le 3 \, W_1(\mathcal{D}, \mathcal{D}').$$

$\square$

Table 1: $L_1$ distance between the average estimate and the true effect, together with the standard deviation of each estimator over 20 repetitions. $\mathcal{H}_d$ denotes homology dimension $d$, and the scale is in 1e-05 for SARS-CoV-2 and ORBIT and 1e-02 for GEOM-Drugs.

| | SARS-CoV-2 | | GEOM-Drugs | | | | ORBIT | | | |
| | $\mathcal{H}_0$ | | $\mathcal{H}_0$ | | $\mathcal{H}_1$ | | $\mathcal{H}_0$ | | $\mathcal{H}_1$ | |
| | $L_1$ dist. | Std. | $L_1$ dist. | Std. | $L_1$ dist. | Std. | $L_1$ dist. | Std. | $L_1$ dist. | Std. |
|---|---|---|---|---|---|---|---|---|---|---|
| PI | 4.544 | 2.712 | 3.215 | 0.740 | 7.041 | 1.283 | 0.571 | 0.530 | 36.063 | 37.704 |
| IPW | 2.936 | 9.035 | **1.670** | 5.756 | 18.886 | 8.337 | **0.073** | 3.722 | 18.210 | 93.978 |
| AIPW | **1.246** | 3.099 | 3.296 | 1.470 | **2.136** | 1.553 | 0.121 | 0.802 | **6.920** | 47.127 |

## C.6 PROOF OF COROLLARY 5.4

We provide only a brief sketch, since the argument follows from standard empirical-process machinery (e.g., Van Der Vaart & Wellner, 1996). By Theorem 5.2, $\sqrt{n}\,(\widehat{\psi}_{\mathrm{AIPW},d} - \psi_d) \rightsquigarrow \mathbb{G}_d$ in $\ell^\infty(\mathbb{T})$, and the continuous mapping theorem applied to $f \mapsto \|f\|_\infty$ implies $T_n \rightsquigarrow \|\mathbb{G}_d\|_\infty$ under $H_0$. If the multiplier bootstrap consistently approximates, conditionally on the data, the law of $\|\mathbb{G}_d\|_\infty$, then its $(1-\alpha)$-quantile $c_{1-\alpha}$ yields an asymptotically valid level-$\alpha$ test; under any fixed alternative with $\|\psi_d\|_\infty > 0$, we have $T_n \geq \sqrt{n}\|\psi_d\|_\infty + o_{\mathbb{P}}(\sqrt{n}) \to \infty$, so the power converges to one.

## D EXPERIMENT DETAILS

We perform experiments on three benchmark datasets: the SARS-CoV-2 CT-scan image dataset, the GEOM-Drugs molecular graph dataset, and the ORBIT point cloud dataset. In all experiments, we construct a synthetic counterfactual dataset $\{(X_i, A_i, Y_i^0, Y_i^1)\}_{i=1}^n$ to facilitate the evaluation of estimators against a known true effect. We begin by randomly selecting and pairing two data samples to form each potential outcome pair, assigning one to $Y^0$ and the other to $Y^1$ in a manner that induces a clear topological contrast. Next, we generate the covariates $X$ and treatment $A$ according to a stochastic data-generating process and treatment assignment mechanism. The same procedure is identically applied to all experiments, with details outlined below.

**Data-generating process.** We assume a setting in which a subgroup structure is imposed on the covariates $X \in \mathbb{R}^5$. The covariates are generated from two subgroups, each governed by a multivariate Gaussian distribution with distinct mean vectors $\mu_1, \mu_2$, and a common covariance matrix $\Sigma$. The covariance matrix $\Sigma$ is set as a diagonal matrix with diagonal values $0.5$, and the mean vector of each subgroup is specified as $\mu_1 = [1, 0.6, -0.7, 2.2, -1]^T$ and $\mu_2 = [0.4, -0.4, -0.6, 3.3, 3]^T$. Covariates for half of the samples are generated from $N(\mu_1, \Sigma)$, while the remaining half are generated from $N(\mu_2, \Sigma)$.

**Treatment mechanism.** Given the covariates $X$, treatment $A \in \{0, 1\}$ is assigned with probability $\pi(X) = expit(-0.5X_1 - 0.1X_2 + 0.6X_3 + 0.1X_4 + 0.1X_5 + 0.5X_2X_3 - 0.7X_1X_3)$. This treatment mechanism is designed such that one subgroup has a higher probability of receiving treatment than the other (see Figure 6a). Upon treatment assignment, the observed outcome is given by $Y = AY^1 + (1 - A)Y^0$.

The aforementioned procedure is repeated 20 times to generate multiple datasets with different realizations of $X$ and $A$, for each of which the PI, IPW, and AIPW estimators are computed. The estimators are constructed by modeling the silhouette regression function $\mu_a$ with function-on-scalar regression employing a Fourier basis expansion, and estimating the propensity score $\pi$ using a random forest classifier. To assess the performance of each estimator, we examine the pointwise mean and the pointwise 1-standard deviation error bands computed across the 20 runs. In addition, we report the $L_1$ distance between the average estimated function and the true topological effect function as a single number summary of estimator accuracy in Table 1. For completeness, we also provide the standard deviation of each estimator in Table 1, obtained by first averaging the empirical covariance matrix to produce a scalar summary of variability, and then taking its square root.

Table 2: The runtime of computing weighted silhouettes for each dataset in seconds.

| Data (Sample Size) | SARS-CoV-2 (2481) | GEOM-Drugs (30433) | ORBIT (2000) |
|---|---|---|---|
| Runtime | 244 seconds | 21 second | 25 seconds |

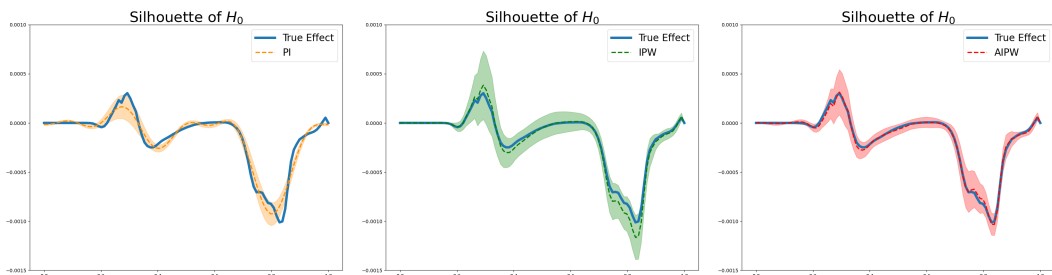

Figure 5: Visualization of the point-wise mean (dotted line) and the point-wise 1-standard deviation error bands (shaded area) of PI, IPW, and AIPW estimators on the SARS-CoV-2 dataset. The true topological causal effect is shown as a blue line.

### D.1 IMPLEMENTATION CONSIDERATIONS

Before discussing the details of each experiment, we first outline several practical considerations relevant to implementing our method.

**Sample Splitting.** As discussed in Section 4, sample splitting enables the use of arbitrary complex nuisance estimators. Thus, the use of sample splitting is crucial for proper implementation, and we illustrate how it can be carried out in practice. Let $D = \{Z_i = (X_i, A_i, Y_i)\}_{i=1}^n$ denote an i.i.d. observed sample of size $n$. We randomly split $D$ into two independent samples $D_1$ and $D_2$, each of size $n/2$. First, we use $D_1$ to fit the nuisance estimators $\hat{\eta}$, and use $D_2$ to compute $\hat{\psi}$ for the estimation of $\psi$. Then, we swap the samples and use $D_2$ for training and $D_1$ for estimation. If we average the two resulting estimators $\hat{\psi}$, we recover full-sample efficiency.

**Computational Cost.** Our framework relies on persistent homology, which incurs substantial computational cost; the *standard* algorithm for persistent homology induces $O(N^3)$ complexity (Edelsbrunner & Morozov), with $N$ denoting the number of simplices. To partially mitigate this computational overhead and improve scalability, several algorithmic techniques have been developed. For instance, Chen & Kerber (2011) proposed an output-sensitive algorithm for persistent homology with $O(N^2 \log^3 N \log \log N)$ complexity. For many applications where 0-dimensional persistent homology suffices, such as our SARS-CoV-2 experiment, additional scalability can be achieved with efficient algorithms that run at $O(N \log N)$ time (Edelsbrunner & Morozov). Nevertheless, even with improved algorithms, our approach utilizing persistent homology may be computationally demanding and provide restricted scalability on large-scale datasets, as this limitation is intrinsic to persistent homology itself. To achieve enhanced scalability, one may consider extending our framework to topological descriptors based on the efficient Euler characteristic curve, although Euler characteristic curve-based descriptors generally lack the expressivity of persistent homology-based methods.

To illustrate that our method is entirely feasible for datasets that are not excessively large, we provide empirical runtime measurements for computing weighted silhouettes across all datasets used in our study. For the SARS-COV-2 dataset, we compute weighted silhouettes on CT images of 1252 infected and 1229 non-infected patients. The GEOM-Drugs is computed on the full dataset of 30422 graphs. For the ORBIT dataset, weighted silhouettes are computed for 1000 pairs of $(Y^1, Y^0)$, amounting to 2,000 silhouettes overall. The GUDHI (The GUDHI Project, 2021) library is used for silhouette computation on an Apple M2 processor, and the runtime results are presented in Table 2.

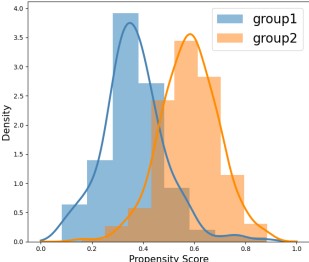
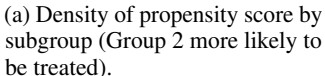
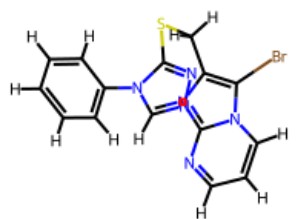
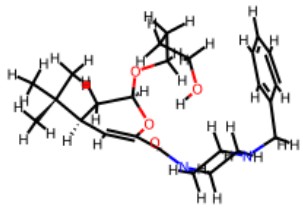

(a) Density of propensity score by subgroup (Group 2 more likely to be treated).

(b) Sample molecular graph A (Axelrod & Gomez-Bombarelli, 2022).

(c) Sample molecular graph B (Axelrod & Gomez-Bombarelli, 2022).

Figure 6: (a) Propensity score distribution by subgroup; (b–c) representative molecular graphs.

## D.2 SARS-CoV-2

The SARS-CoV-2 dataset contains CT-scans collected from real patients in Brazil, who are infected or non-infected by COVID-19. In our experiment, 500 potential outcomes pairs $(Y^0, Y^1)$ are constructed according to the following procedure: (i) 500 infected images are sampled and assigned to $Y^0$, (ii) 375 non-infected and 125 infected images are sampled and assigned to $Y^1$, (iii) $Y^0$ and $Y^1$ are randomly paired. By experimental design, TATE exhibits treatment effect since 75% of $Y^1$ is non-infected where as every individual in $Y^0$ is infected. Silhouettes are computed using sublevel set filtration on a filtered cubical complex (see Appendix A) with $r = 0.1$. For the nuisance estimators, function-on-scalar regression is implemented with 10 basis functions, while random forest is constructed with 100 trees. The point-wise 1-standard deviation error bands of the respective estimators are demonstrated in Figure 5, and see Table 1 for numerical summaries of estimator performance.

## D.3 GEOM-DRUGS

The GEOM-Drugs dataset consists of graph-structured representations of molecular compounds, as shown in Figure 6b and 6c. All graph nodes have six features, and we assign each node a scalar weight as a randomly sampled convex combination of these feature values. Each edge is assigned a weight given by the maximum of its adjacent node weights. Since 1-dimensional homology features have infinite death times, we replace them with values sampled uniformly from $[6.5, , 8.5]$. In our experiment, 1000 potential outcomes pairs $(Y^0, Y^1)$ are constructed according to the following procedure: (i) randomly sample 1000 graphs with one loop and assign them to $Y^0$, (ii) randomly sample 750 graphs with two loops and 250 graphs with one loop and assign them to $Y^1$. This allocation scheme gives rise to significant topological differences between the counterfactual groups. We compute silhouettes with $r = 1$, while the nuisance estimators for $\mu_a$ and $\pi$ are specified with 5 basis functions and 100 trees, respectively. Figures 7 provides a visualization of the point-wise mean and the point-wise 1-standard deviation error bands of the estimators per homology dimension, and see Table 1 for numerical summaries of estimator performance.

## D.4 ORBIT

The ORBIT dataset (Adams et al., 2017) is a synthetic point cloud dataset that is generated by simulating different dynamical systems characterized by a parameter $s$. Given a random initial point $(x_0, y_0) \in [0, 1]^2$ and $s > 0$, we generate point clouds consisting of 1000 points as follows:

$$x_{n+1} = x_n + s y_n (1 - y_n) \mod 1$$
$$y_{n+1} = y_n + s x_n (1 - x_n) \mod 1,$$

In this experiment, we use $s = 3.5, 4, 4.1$ to generate 1000 samples for each value of $s$, with the resulting point clouds illustrated in Figure 2a: $s = 3.5$ (top right), $s = 4$ (bottom), $s = 4.1$ (top left). From each triplet of point clouds generated by $s = 3.5, 4, 4.1$, we randomly select two point clouds and assign the one corresponding to the higher $s$ value as $Y^1$ with probability 0.7. This procedure

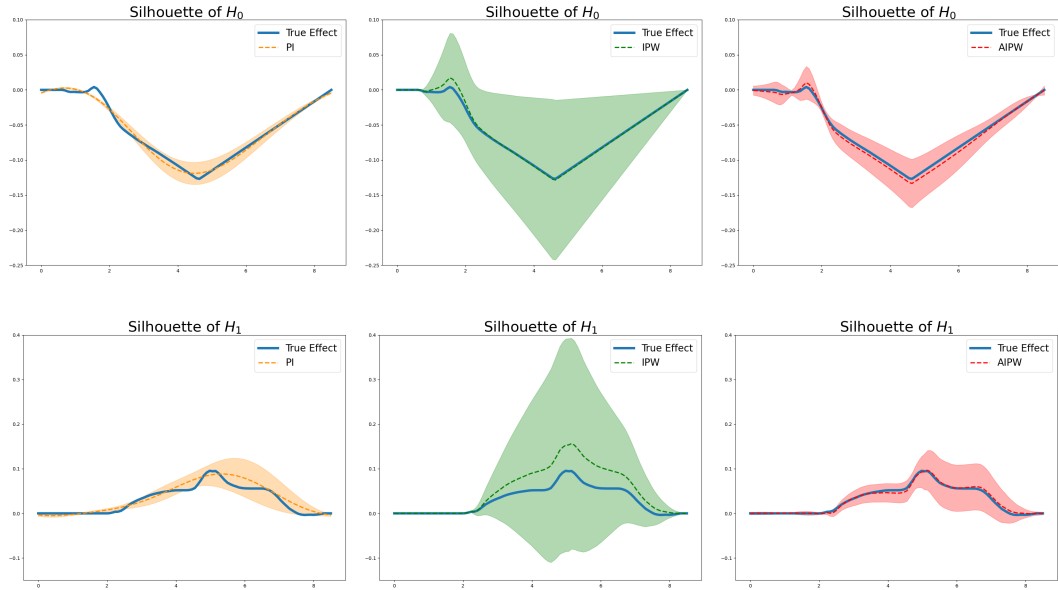

Figure 7: Visualization of the point-wise mean (dotted line) and the point-wise 1-standard deviation error bands (shaded area) of 0-dimensional (top) and 1-dimensional (bottom) PI, IPW, and AIPW estimators on the GEOM-Drugs dataset. The true topological causal effect is shown as a blue line.

yields pairs of matched potential outcomes $(Y^0, Y^1)$ for all 1000 samples, where the treated potential outcome is intentionally designed to possess more pronounced topological features. Here, we use 3 bases to model the silhouette regression function and 100 trees to estimate the propensity score, and the silhouettes are computed using Alpha filtration (see Appendix A) with power weight $r = 3$.

**Results.** Figure 8 illustrates the true target silhouettes in homology dimensions 0 and 1, which clearly demonstrate a causal treatment effect on first-order homological features of point clouds. In particular, the positive values of the 1-dimensional silhouette indicate the emergence of holes in the treated point cloud. Our aim is to recover the magnitude of this silhouette function, as larger values correspond to more substantial structural changes. Consistent with previous results, the IPW estimator tends to overestimate the treatment effect, whereas the plug-in estimator underestimates it. The AIPW estimator, by contrast, yields a substantially more accurate estimate of the true silhouette function. Moreover, the near-zero silhouette for 0-dimensional homology features suggests that the data's 0-dimensional homology, including connected components, remained largely unchanged after treatment. Figure 9 illustrates the pointwise mean and the pointwise 1-standard deviation error bands for each of the PI, IPW, and AIPW estimators on the ORBIT dataset, and see Table 1 for numerical summaries of estimator performance.

**Hypothesis Testing.** The ORBIT dataset provides an appropriate setting for empirically evaluating the hypothesis test presented in Corollary 5.4, given that the 0-dimensional silhouette exhibits no discernible topological effect, whereas the 1-dimensional silhouette displays significant topological effect (Figure 8). Accordingly, we conducted the hypothesis test of null $H_0 : W_1(\mathcal{D}^1, \mathcal{D}^0) = 0$ to examine whether the test outcomes are consistent with our visual interpretation of Figure 8. We generate 1000 bootstrap samples $\hat{\mathbb{G}}_n$ and use $\alpha = 0.05$. For 0-dimensional homology, the test statistic $T_n = 0.0037$, whereas the $1 - \alpha$ quantile of $\|\hat{\mathbb{G}}_n\|_\infty$ is 0.0277. Thus, consistent with our expectation, we cannot reject the null hypothesis. For 1-dimensional homology, the test statistic $T_n = 0.4242$, whereas the $1 - \alpha$ quantile of $\|\hat{\mathbb{G}}_n\|_\infty$ is 0.1107. As expected, the test rejects the null hypothesis, indicating a significant treatment-induced topological shift in the 1-dimensional homology of the data.

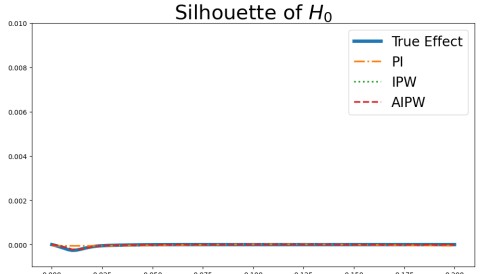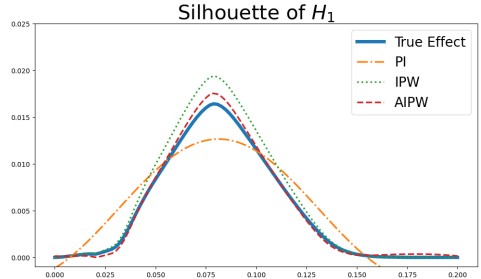

Figure 8: 0-dimensional (left) and 1-dimensional (right) true silhouette functions and its PI, IPW, AIPW estimates for the ORBIT dataset.

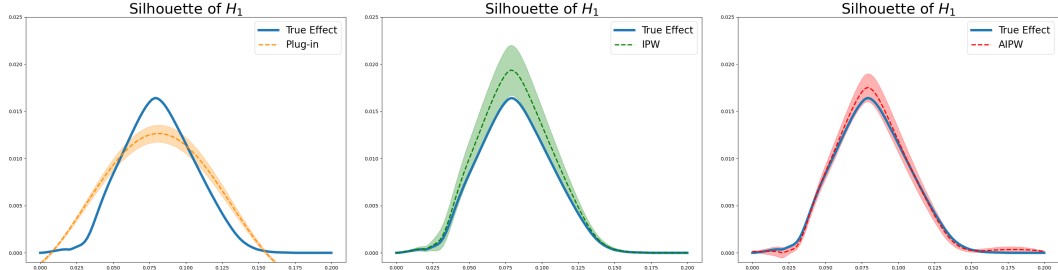

Figure 9: Visualization of the pointwise mean (dotted line) and the pointwise 1-standard deviation error bands (shaded area) of 1-dimensional PI, IPW, and AIPW estimators on the ORBIT dataset. The true topological causal effect is shown as a blue line.

## D.5 ESTIMATION UNDER MODEL MISSPECIFICATION

In previous experiments, complex nonparametric models were used to flexibly estimate the nuisance functions. In this section, we investigate how the estimators perform under severe model misspecification of nuisance functions. Specifically, we are interested in observing how the performance of estimators varies across the following scenarios: (i) $\pi$ is severely misspecified and $\mu$ is the correct model, and (ii) $\mu$ is severely misspecified and $\pi$ is the correct model.

Since we have knowledge of the true treatment process, the correct $\pi$ is modeled as $\hat{\pi}(X) = expit(\beta_1 X_1 + \beta_2 X_2 + \beta_3 X_3 + \beta_4 X_4 + \beta_5 X_5 + \beta_6 X_2 X_3 - \beta_7 X_1 X_3)$, whereas $\hat{\pi}(X) = expit(\beta_1 X_1 + \beta_2 X_3)$ is used for the misspecified model. For the outcome regression $\mu$, whose true form is unknown, we retain the function-on-scalar regression using Fourier basis expansion. We choose a sufficiently large number of basis to represent the correctly specified model and a deliberately reduced number of basis to construct the misspecified model. Specifically, for the SARS-CoV-2, GEOM-Drugs, and ORBIT datasets, we respectively use 30, 10, and 5 bases for the correct regression model, and 7, 2, and 2 bases for the misspecified regression model. Figures 10, 11, and 12 illustrate the visualized results, and Table 3 provides quantitative summaries of estimator performance. We can observe that across all datasets and misspecification settings, the AIPW estimator generally has the best performance.

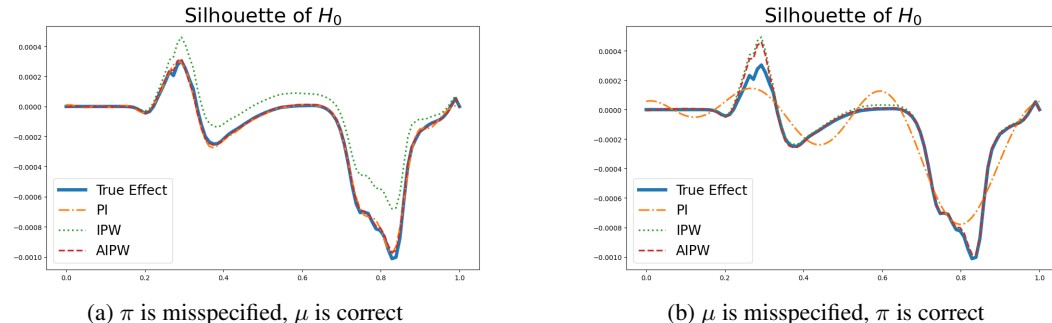

(a) $\pi$ is misspecified, $\mu$ is correct

(b) $\mu$ is misspecified, $\pi$ is correct

Figure 10: True silhouette and its PI, IPW, AIPW estimates for 0-dimensional homological features on the SARS-CoV-2 dataset when one of the nuisance functions is misspecified.

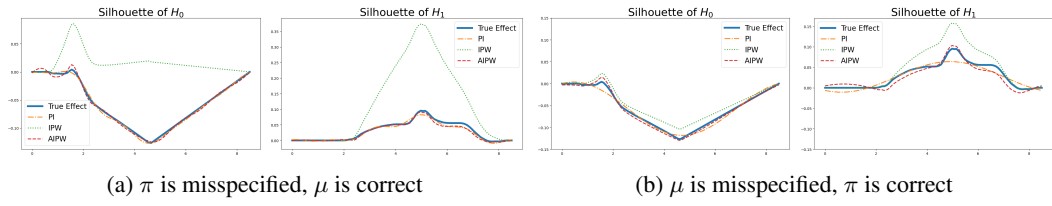

(a) $\pi$ is misspecified, $\mu$ is correct

(b) $\mu$ is misspecified, $\pi$ is correct

Figure 11: True silhouette and its PI, IPW, AIPW estimates for 0- and 1-dimensional homolgical features on the GEOM-Drugs dataset when one of the nuisance functions is misspecified.

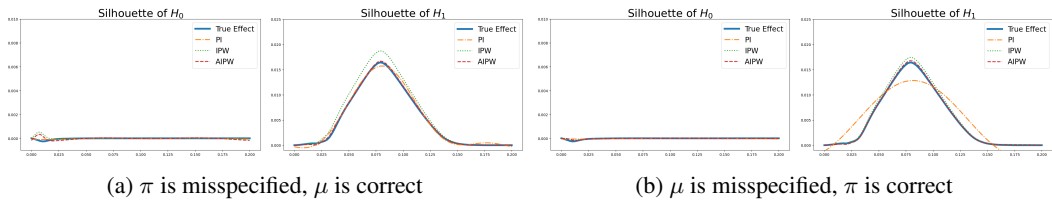

(a) $\pi$ is misspecified, $\mu$ is correct

(b) $\mu$ is misspecified, $\pi$ is correct

Figure 12: True silhouette and its PI, IPW, AIPW estimates for 0- and 1-dimensional homolgical features on the ORBIT dataset when one of the nuisance functions is misspecified.

Table 3: $L_1$ distance between the average estimate and the true effect, together with the standard deviation of each estimator over 20 repetitions, when either of the nuisance functions is misspecified. $\mathcal{H}_d$ denotes homology dimension $d$, and the scale is in 1e-05 for SARS-CoV-2 and ORBIT and 1e-02 for GEOM-Drugs.

| **SARS-CoV-2** ($\mathcal{H}_0$) | $\pi$ **is misspecified,** $\mu$ **is correct** | | $\mu$ **is misspecified,** $\pi$ **is correct** | |
|---|---|---|---|---|
| | $L_1$ dist. | Std. | $L_1$ dist. | Std. |
| PI | 1.390 | 3.132 | 7.985 | 3.132 |
| IPW | 9.674 | 4.421 | 2.243 | 4.328 |
| AIPW | **0.786** | 2.761 | **1.529** | 3.395 |

| **GEOM-Drugs** | $\pi$ **is misspecified,** $\mu$ **is correct** | | | | $\mu$ **is misspecified,** $\pi$ **is correct** | | | |
|---|---|---|---|---|---|---|---|---|
| | $\mathcal{H}_0$ | | $\mathcal{H}_1$ | | $\mathcal{H}_0$ | | $\mathcal{H}_1$ | |
| | $L_1$ dist. | Std. | $L_1$ dist. | Std. | $L_1$ dist. | Std. | $L_1$ dist. | Std. |
| PI | **2.006** | 1.133 | 4.022 | 0.865 | 4.376 | 1.133 | 6.879 | 0.865 |
| IPW | 61.796 | 3.602 | 77.963 | 4.426 | 10.299 | 2.831 | 16.245 | 3.564 |
| AIPW | 2.343 | 1.168 | **3.432** | 0.897 | **3.380** | 1.390 | **6.842** | 1.112 |

| **ORBIT** | $\pi$ **is misspecified,** $\mu$ **is correct** | | | | $\mu$ **is misspecified,** $\pi$ **is correct** | | | |
|---|---|---|---|---|---|---|---|---|
| | $\mathcal{H}_0$ | | $\mathcal{H}_1$ | | $\mathcal{H}_0$ | | $\mathcal{H}_1$ | |
| | $L_1$ dist. | Std. | $L_1$ dist. | Std. | $L_1$ dist. | Std. | $L_1$ dist. | Std. |
| PI | **0.514** | 0.688 | 7.298 | 37.179 | 0.600 | 0.688 | 36.323 | 37.179 |
| IPW | 0.766 | 2.300 | 17.576 | 45.287 | **0.156** | 1.482 | 7.092 | 55.763 |
| AIPW | 1.347 | 0.681 | **2.605** | 39.582 | 0.284 | 0.748 | **2.354** | 40.813 |

