# OpenReview forum: "Topological Causal Effects"
_ICLR.cc/2026/Conference — ICLR 2026 Poster_

### Official Review · Reviewer_XnWi · 2025-10-24

**Soundness:** 3
**Presentation:** 4
**Contribution:** 3
**Rating:** 8
**Confidence:** 3

**Summary:**

This work introduces a novel framework for topological causal inference, combining the Potential Outcome framework with Topological Data Analysis tools in order to estimate causal effects when the outcome lies outside Euclidean spaces. The authors detail the identification and estimation steps and provides interesting real-world experiments.

**Strengths:**

This paper is strong, all claims are well explained and supported by theoretical and experimental results. All proofs are left in the appendix which allows for an easy read. This work seems quite novel and very useful as multiple real world examples are provided.

**Weaknesses:**

The biggest weakness in my opinion lies in the strength of the assumptions required.

**Questions:**

* I believe the assumptions to be quite strong. Could you explain how one can test these assumptions. Are these assumptions satisfied in the real-world examples provided?
 * There exists two main causal frameworks: The potential outcome framework and Pearl's DAG framework. I wonder what arguments motivated the authors to chose the potential outcome framework? Is this work compatible with the DAG framework?

---

> ### Author Response · Authors · 2025-11-22
>
> We express our sincere gratitude to the reviewer for appreciating our work, and address the questions below.
>
> 1. **Strength of assumptions.** We are grateful for the reviewer's comment on this. There are two types of assumptions in our work.
>     - (C1)–(C3) are standard identification assumptions in causal inference, and our framework inherits them directly. As in all observational studies, the key requirement is the no-unmeasured-confounding assumption (C2), which cannot be tested from observed data alone. However, standard sensitivity analyses can be used to assess how violations of this assumption would affect conclusions. In our setting, the consistency assumption (C1) appears slightly different because the outcome is a filtration rather than a scalar, but it remains a mild requirement: it holds whenever the observed outcome satisfies $Y = Y^a$ for $A=a$, and the filtration function applied to construct topological summaries is the same for both potential outcomes.
>     - For inference, we rely on the technical conditions (A1)–(A5), which are standard and commonly employed in semiparametric, efficiency-theory–based causal inference (e.g., [1,2] (Kennedy 2016, 2024)). They likewise parallel the assumptions employed in recent developments in semiparametric functional causal inference (e.g., [3] (Testa et al., 2025)). We also note that a slightly weaker condition, (A5'), can replace (A5) in many cases, as discussed in Section 5. Overall, the assumptions (A1)–(A5) are not novel or unusually restrictive, and each has clear precedent in the existing literature. We agree that some of these assumptions are difficult to test directly from observed data. However, they can typically be justified a priori using established results in the nonparametric estimation literature. For a given class of nonparametric models, one can appeal to known guarantees on convergence rates, consistency, and uniform Lipschitz behavior (e.g., [4] (Wasserman, 2006); [5] (Györfi, 2013)), which provide theoretical support for assumptions such as (A3)–(A5). Finally, Assumption (A6), used to derive the stability bound in Theorem 5.4, is very mild: it only requires the underlying persistence diagrams to be bounded, which is equivalent to assuming the input data lie within a finite radius. Under this boundedness, the existence of a global constant $L$ follows immediately.
>
> 2. **Choice of framework** We chose the potential outcome framework because it provides a direct, transparent definition of our causal estimand building upon functional analysis tools: the topological average treatment effect is simply the contrast between the silhouette functions of the potential outcomes $Y^1$ and $Y^0$. This framework allows us to utilize rigorous semiparametric inference tools, which are crucial for deriving the second-order bias properties of our estimator and for constructing valid confidence bands.
>
>     Our approach is broadly compatible with Pearl's DAG framework in the sense that a DAG can encode the same identifying assumptions, most importantly, the conditional independence $A \perp Y^a \mid X$. However, extending our results directly through the DAG formalism would introduce additional complexities: causal effects on objects in topological spaces are not straightforward to define using standard structural equation models, and deriving influence-function–based inference in that setting is considerably more involved. For these reasons, the potential outcome framework provides a cleaner and more tractable foundation for defining the estimand and conducting inference, while remaining conceptually consistent with causal models expressible in the DAG framework.
>
> We wish that our response has adequately addressed your questions and concerns. Should any concerns remain, we would be more than happy to provide further clarification, and we again thank the reviewer for their time and input.
>
>
> [1] Edward H Kennedy. Semiparametric theory and empirical processes in causal inference. Statistical causal inferences and their applications in public health research, pp. 141–167, 2016.
>
> [2] Edward H Kennedy. Semiparametric doubly robust targeted double machine learning: a review. Handbook of Statistical Methods for Precision Medicine, pp. 207–236, 2024.
>
> [3] Lorenzo Testa, Tobia Boschi, Francesca Chiaromonte, Edward H Kennedy, and Matthew Reimherr. Doubly-robust functional average treatment effect estimation. arXiv preprint arXiv:2501.06024, 2025.
>
> [4] Wasserman, L. All of nonparametric statistics. New York, NY: Springer New York. 2006
>
> [5] Györfi, L. Nonparametric statistics. 2013.

---

### Official Review · Reviewer_Zney · 2025-10-29

**Soundness:** 4
**Presentation:** 3
**Contribution:** 4
**Rating:** 6
**Confidence:** 3

**Summary:**

This work proposes a novel notion of treatment effect for causal inference, defined through algebraic topology.
Given unstructured or high-dimensional treatment outcomes, the idea is to apply a dimensionality reduction approach from
topological data analysis to extract homology features that represent the underlying topological structure of the data.
This allows comparing potential outcomes before and after an intervention from differences in these features.
To this end, the approach constructs persistence diagrams from the homology features (e.g., Fig 1 right) and
compares their expected difference in silhouette score, resulting in the topological treatment effect measure (Eq. 3).

The remainder of the work considers estimation, asymptotic properties, and empirical study of the topological TE. They study the plug-in (PI) and inverse-probability-weighted estimators (IPW and AIPW), show under which conditions they are unbiased
and how to construct confidence bands, and derive a test for the null hypothesis of no topological TE.
Evaluations cover three benchmarks derived from real-world datasets, and show that in particular the AIPW gives a close approximation of the true silhouette function.

**Strengths:**

The connection between algebraic topology and causal inference is original and has a clear practical value,
given that it allows addressing high-dimensional and/or structurally complex outcomes, for which few methods exist.
The presentation is clear and provides example illustrations of the important concepts. There is a careful theoretical analysis with independently interesting results, such as the stability bounds for weighted silhouettes, and the experiments match the theoretical results and are convincing.
Overall, studying a topology-aware variant of treatment effect is a worthwhile contribution and could open an interesting future work direction integrating insights from TDA to causal inference.

**Weaknesses:**

The experimental evaluation would be stronger if a comparison to existing TE estimators were given if applicable, and the related work could be expanded with references to previous methods (esp. addressing a high-dimensional setting). Furthermore, while some comments on practical considerations are given (ln. 256-), the authors might consider including a section on implementation considerations and details in the main text or at least Appendix.

**Questions:**

- Related to the above weakness, can an existing TE be included in the experiments, for example to the ORBIT benchmark?
- Given your claim in ln. 354-355 that the hypothesis test (Cor 5.5) is of primary interest to most practitioners, it would be interesting to study it empirically on the datasets.
- Can you comment on the connection between the functional ATE (Testa et al., 24) and the topological one, as alluded in line 192?

---

> ### Author Response · Authors · 2025-11-22
>
> We sincerely thank the reviewers for their time and thoughtful feedback. We would like to address each of your comments below.
>
> 1. **Comparison to existing TE estimators.** We thank the reviewer for raising the question of comparison with existing TE estimators. By “standard TE estimators” one may refer to conventional ATE estimators (e.g., plug-in, IPW, AIPW) applied after first somehow collapsing each persistence diagram to a finite-dimensional Euclidean summary (such as scalar topological summaries, vectorized persistence images, or basis coefficients of silhouette functions). Such procedures estimate $\mathbb{E}[g(Y^1)] - \mathbb{E}[g(Y^0)]$, for some ad hoc summary map $g$ that embeds topological data into $\mathbb{R}^k$, and hence treat the resulting outcomes as ordinary Euclidean objects.
>
>     In contrast, our proposed topological TE is defined directly in the metric space of persistence diagrams via the family of silhouette functions $\psi_d(t) = \mathbb{E}[\phi^1(t; \mathcal{D}_d^1)] - \mathbb{E}[\phi^0(t; \mathcal{D}_d^0)], \ t \in \mathbb T$, and is equipped with Wasserstein–stable, infinite-dimensional inference (Theorems~3.1–5.4). Thus, “TE + vectorization” baselines are **not** targeting the same estimand: they ignore the full path $t \mapsto \psi_d(t)$ and the underlying diagram geometry, while our framework is explicitly topology-aware. Therefore, while such vectorization-based TE estimators can be used as naive baselines, they are conceptually distinct from, and generally less expressive than, the topological causal effect proposed in our work. We have clarified this point in the revised manuscript (lines 206-211).
>
> 2. **Implementation considerations.** We appreciate the reviewer's thoughtful suggestion. Additional comments on implementation considerations would indeed help clarify the details of our method, and we have added Appendix D.1 in the updated manuscript to reflect this feedback accordingly.
>
> 3. **Empirical hypothesis test.** Thank you for the insightful suggestion. We agree that including an empirical hypothesis test would strengthen our manuscript. To this end, the ORBIT dataset provides an appropriate setting for hypothesis testing, given that the 0-dimensional silhouette exhibits no discernible topological effect, whereas the 1-dimensional silhouette displays significant topological effect (Figure 11). Accordingly, we conducted hypothesis testing on the ORBIT dataset, and the corresponding results are reported in Appendix D.4.
>
> 4. **Connection to functional ATE** The connection to the functional ATE (FATE) of [1] (Testa et al., 2025) arises because our target estimand, TATE is itself a function valued in a Hilbert space. This places TATE within the broader class of functional causal estimands and allows us to leverage tools from functional causal inference. However, the role of TATE is fundamentally different from a general FATE: it is a topology-aware functional estimand, where each value $\phi(t;\\mathcal{D}_d)$ encodes the homological structure of the data at scale $t$. Thus, TATE captures treatment-induced changes in the topology of the underlying object (graph, point cloud, image, etc.), rather than changes in an arbitrary functional outcome.
>
>     This distinction enables methodological developments that are not available in standard FATE frameworks. In particular, combining our silhouette-based representation with the new stability bound in Theorem 5.4 allows us to formulate a formal test of $ H_0: W_1(\mathcal{D}^1,\mathcal{D}^0)=0$,
>     i.e., no topological shift between potential outcomes (Corollary 5.5). Moreover, because silhouettes are 1-Lipschitz (Lemma 2.1), we establish weak convergence under milder conditions than those typically required in functional causal inference.
>
>     Thus, while TATE fits naturally within the FATE paradigm, it extends it to settings with non-Euclidean, topologically structured outcomes, enabling causal analysis and hypothesis testing in domains where classical FATE is not directly applicable. We clarify this distinction in the revised manuscript (lines 186-191).
>
> We hope that the above response sufficiently resolves your questions, and should any concerns remain, we would be happy to provide further clarifications.
>
> [1] Lorenzo Testa, Tobia Boschi, Francesca Chiaromonte, Edward H Kennedy, and Matthew Reimherr. Doubly-robust functional average treatment effect estimation. arXiv preprint arXiv:2501.06024, 2025.

---

### Official Review · Reviewer_ybeB · 2025-10-31

**Soundness:** 3
**Presentation:** 2
**Contribution:** 3
**Rating:** 6
**Confidence:** 3

**Summary:**

The paper introduces a framework for topological causal inference that defines treatment effects through changes in the topology of subjects. The authors utilize persistent homology from TDA to capture structural shifts due to intervention by a treatment, proposing topological average treatment effect defined as expected differences in power-weighted silhouettes of persistence diagrams between treated and control groups. They develop estimators, derive asymptotic properties, establish stability bounds for weighted silhouettes, and provide a formal hypothesis test for the null of no topological effect. Experiments on semi-synthetic datasets demonstrate the method's effectiveness in capturing topological causal effects.

**Strengths:**

**S1.** The work is novel and makes a foundational contribution that meaningfully extends causal inference to the topology of treatment units, addressing an important gap where structural changes encode scientifically relevant effects.

**S2.** The authors propose a solid theoretical framework with comprehensive analysis.

**S3.** The paper introduces a hypothesis test for no topological effect with consistency guarantees.

**S4.** Experimental validation demonstrates effectiveness on semi-synthetic data.

**Weaknesses:**

**W1.** The authors do not discuss scalability and computational complexity in the main paper. This is mentioned briefly in the Discussion section as a limitation, but given the claim for practical relevance, I believe this needs to be discussed in the paper.

**W2.** The silhouette is an aggregated representation of topological features, which could potentially obscure the change in each homological feature. Ideally this should also be discussed further in the paper.

**W3.** The weighted silhouette depends on the hyperparameter $r$. There is lack of guidance in the paper on how this is/should be selected, and how it influences the results.

**W4.** While the paper is indeed novel, it is not the first bridge TDA and causal inference. To my knowledge, a recent work [1] connects the two. [1] is fairly recent and it is understandable if the authors have missed it in their literature review, but the claim in being the first to bridge TDA with causal inference should be corrected. That being said, the paper is still novel in characterizing topological causal effect.

[1] Farzam, A., Aloui, A., Tarokh, V., & Sapiro, G. Topology-Aware Robust Representation Balancing for Estimating Causal Effects. In High-dimensional Learning Dynamics 2025.

**Questions:**

**Q1.** What is the computational complexity of the proposed method? Can you provide empirical runtime comparisons or discuss scalability strategies for large datasets?

**Q2.** How sensitive are the results to the choice of power parameter $r$ in the weighted silhouette? Are there specific guidelines for selecting this parameter (e.g., for different applications)?

---

> ### Author Response · Authors · 2025-11-22
> **Official Comment by Authors (1/2)**
>
> We sincerely thank the reviewer for the valuable comments and questions, and address each of your major concerns below.
> 1. **W1. & Q1.** We appreciate the reviewer’s insightful comment on computation and scalability, which we likewise recognize as a central challenge for practical deployment. The computational bottleneck of our approach primarily arises from the intensity of persistent homology (PH) computation, where the *standard* algorithm induces $O(N^3)$ complexity (see *column algorithm* in Section 26.4 of [1] (Edelsbrunner and Morozov, 2017)), with $N$ denoting the number of simplices. However, several algorithmic techniques have been developed to partially mitigate this computational overhead, enabling improved scalability; for instance, [2] (Chen et al., 2013) demonstrates $O(N^2 \log^{3}N \log\log N)$ complexity for PH computation. For many applications where 0-dimensional PH is sufficient, such as our SARS-CoV-2 experiment, additional scalability can be achieved with efficient algorithms that run at $O(N \log N)$ time (see *fast algorithm for 0-dimensional persistence* in Section 26.4 of [1] (Edelsbrunner and Morozov, 2017)). To mitigate the computational burden intrinsic to PH computation, one may instead adopt more efficient topological summaries, such as Euler characteristic curves, as a practical alternative (see lines 511–513). We include this discussion in Appendix D.1 of our updated manuscript. To illustrate that our method is entirely feasible for datasets that are not excessively large, we have additionally included the empirical runtime results in Table 2 of Appendix D.
>
> 2. **W2.** We thank the reviewer for this insightful clarification. As the reviewer has pointed out, the aggregated representation of silhouettes may obscure changes in individual homological features (lines 201-203, 508-509). If one wishes to detect changes in individual homological features, our framework can be directly extended to persistent landscapes (PL), which yield a finer topological resolution (lines 203-205, 509-511). Although we use silhouettes rather than PL because silhouettes (i) provide a concise and unified summary of the *total* topological variation, and (ii) allow weighting schemes that highlight more meaningful features, our framework readily extends to PL in settings where changes in individual or specific homological features are of primary interest.
>
> 3. **W3. & Q2.** We appreciate the reviewer’s question regarding the choice of the power parameter $r$. In our framework, all identification and asymptotic results are derived for an arbitrary fixed $r>0$; that is, Theorems 5.1-5.4 and Corollary 5.5 remain valid uniformly over any fixed choice of $r$ (lines 130-132). Thus, $r$ does not affect the validity or performance of the proposed estimation procedure, but rather controls which topological features are emphasized: larger $r$ upweights long-lived (high-persistence) features, while smaller $r$ gives relatively more weight to short-lived features (lines 129-130). Therefore, $r$ controls the trade-off between sensitivity to fine-scale versus macroscopic topological structure.
>
>     In practice, we view the power parameter $r$ as a problem-dependent, user-specified tuning choice. For instance, in our SARS-CoV-2 experiment, the persistence diagrams of non-infected and infected patients exhibited distinct concentrations of low-persistence features near the diagonal (Figure 3-(b)). To better reflect these differences in the silhouettes, we selected a small value $r = 0.1$ to up-weight less persistent features. Empirically, however, we observed that, over a broad range of $r$-values, the qualitative shape and statistical significance of the estimated topological effects remained stable, with only minor changes in amplitude, suggesting that our main conclusions are not overly sensitive to the specific choice of $r$. For practitioners, we recommend either choosing $r$ using domain knowledge (e.g., employing larger $r$ to focus on dominant long-lived topological features) or via a simple validation-criterion or cross-validation step applied within a single treatment arm. We have added this guidance in Remark 1 of the revised manuscript.

---

> > ### Author Response · Authors · 2025-11-22
> > **Official Comment by Authors (2/2)**
> >
> > 4. **W4.** We appreciate the reviewer for pointing us to Farzam et al. (2025). We have now added this work to the related literature. However, the contribution of Farzam et al. (2025) is conceptually distinct from ours. Their method uses topological information as a regularizer to improve representation-balancing neural networks for conventional ATE estimation on Euclidean outcomes. In other words, topology is incorporated as an auxiliary feature just to stabilize propensity score estimation, but the causal estimand itself remains the standard scalar ATE.
> >
> >     By contrast, our work is, to the best of our knowledge, the first to formally define and identify a topological causal effect, where the outcome itself lives in a topological space and the causal estimand is a function-valued contrast of persistence-based summaries. Our framework specifies identification, efficient estimation, weak convergence, and a formal hypothesis test for the null of "no topological shift" w.r.t persistence homology. These components do not appear in prior work, including Farzam et al. (2025), which does not define or analyze causal effects on persistent homology or any topological descriptor. We have revised the manuscript to explicitly acknowledge this distinction (lines 46-50).
> >
> > We hope that our response has sufficiently addressed your questions, and we would be glad to address any further clarification requests should any concerns remain.
> >
> > [1] Edelsbrunner, Herbert and Morozov, Dmitriy. 26 PERSISTENT HOMOLOGY
> >
> > [2] Chao Chen and Michael Kerber. An output-sensitive algorithm for persistent homology. Comput. Geom., 46(4):435–447, 2013.

---

### Official Review · Reviewer_R7QM · 2025-11-01

**Soundness:** 2
**Presentation:** 3
**Contribution:** 3
**Rating:** 4
**Confidence:** 3

**Summary:**

This paper bridges topological data analysis and causal inference to estimate causal effects for non-Euclidean outcomes, like molecular graphs and point clouds. Core contributions include defining the Topological Average Treatment Effect as the expected difference in power-weighted silhouettes of persistence diagrams between treatment groups, proposing plug-in, inverse probability weighting, and augmented IPW estimators, deriving theoretical guarantees, including asymptotic distribution, Lipschitz stability, convergence, and hypothesis testing. The performances of estimators are validated on three semi-synthetic/synthetic datasets with visual evidence of AIPW outperforming alternative estimators.

**Strengths:**

The paper’s primary strength lies in its innovative problem formulation: reframing causal effects as treatment-induced topological changes (rather than scalar/vector shifts) and integrating topological data analysis tools to quantify these changes. This opens a new direction for causal inference on non-Euclidean data, with clear relevance to biomedicine, neuroscience, and other fields where structural outcomes matter. Experimental design uses well-characterized semi-synthetic datasets with known true topological effects, providing a controlled setting to test proposed estimators. The paper also effectively breaks down complex topological and causal concepts, with concrete examples for basic definitions and straightforward interpretation of the topological average treatment effect, enhancing accessibility for readers unfamiliar with topological data analysis.

**Weaknesses:**

W1: The paper suffers from inadequate theoretical rigor. It does not provide explicit bias and variance formulations for IPW and AIPW estimators. While claiming AIPW some kind of double robustness benefit? But it offers no verification, for instance, analyses of how misspecified propensity score or outcome prediction models, extreme predictions, or skewed propensity scores impact estimation performance are entirely absent. There is also no discussion of whether robustness holds under realistic conditions like moderate nuisance model misspecification or unmeasured confounders.

W2: Experimental validation is insufficient. Results rely solely on visual comparisons of silhouette curves, with no tabular reporting of quantitative metrics (such as estimated bias, mean squared error, variance, confidence interval, or coverage etc.) for estimation accuracy. Statistical tests to confirm AIPW’s performance gains over other estimators are not discussed, leaving uncertainty about reproducibility or whether results are specific to chosen random seeds.

W3: Critical robustness analyses are omitted. The paper does not evaluate how misspecifying propensity score models (e.g., linear models for non-linear true scores) or outcome prediction models (e.g., incorrect Fourier basis sizes) affects estimator bias and variance. It also fails to test performance under skewed or extreme propensity score distributions to contextualize the proposed framework’s advantages.

**Questions:**

Q1: Can you provide explicit mathematical formulations for the bias and variance of your proposed estimators? How do misspecifications of propensity score or outcome prediction models, or skewed propensity scores, affect these quantities? Then how do you rigorously verify AIPW’s DR property? Do extreme predictions or skewed propensity scores erode double robustness?

Q2: Will you provide tabular experimental results with quantitative metrics for your estimators across datasets? Such as estimated bias, variance, or AUC based on your outcome type? Can you confirm that AIPW’s performance gains over the others are statistically significant? Are your experimental results seed-specific? Please provide pairwise t-test results across multiple random seeds to demonstrate reproducibility.

Q3: How is the comparison between your proposed methods and other existing functional causal inference baselines (such as Testa et al., 2025) in terms of accuracy and robustness?

---

> ### Author Response · Authors · 2025-11-22
> **Official Comment by Authors (1/2)**
>
> We sincerely thank the reviewer for their time and valuable comments. We address each concern in turn below.
>
> 1. **W1. & Q1.** The double-robustness guarantee, or more generally the second–order error, indeed follows from Assumption (A4) and the von Mises expansion in Equation (11). The bias of the proposed AIPW estimator is captured by the second–order remainder term $R_2(\hat{\mathbb{P}}, \mathbb{P})$ in Equation (12). This gives rise to the nonparametric rate condition in Assumption (A4), which underlies both $\sqrt{n}$ consistency and weak convergence to a Gaussian process. For $\sqrt{n}$ consistency it is sufficient that
> $$
> R_2(\hat{\mathbb{P}}, \mathbb{P}) = O\_\mathbb{P}(||\hat{\pi}-\pi||\_{\mathbb{P}, 2} \sum\_{a\in\mathcal A}||\hat\mu_a-\mu_a||\_{\mathbb{P}, 2}) = O\_\mathbb{P}(n^{-1/2})
> $$
> for every $t \in \mathbb{T}$. In other words, we attain root-$n$ rates whenever the product of the two nuisance errors is of order $n^{-1/2}$, even if each nuisance is itself estimated at a slower nonparametric rate (e.g. $n^{1/4}$). This is precisely the formal statement of double robustness in the nonparametric setting. In the classical parametric setting, this condition implies that correct specification of "either" nuisance function is sufficient. The asymptotic variance is given by $\text{var}[\varphi(t,Z;\eta)]$, where $\varphi$ is specified in Equation (8).
>
>     In contrast, the PI and IPW estimators have *first-order* bias terms depending solely on their own nuisance errors (e.g. $||\hat{\mu}-\mu\||$ or $||\hat{\pi}-\pi||$), so they cannot achieve root-$n$ rates unless the corresponding nuisance is estimated exactly at that rate. We now highlight this distinction clearly in the manuscript (lines 359-365).
>
>     Plus, in Appendix B of the revised manuscript, we now provide an explicit analysis of the bias and variance under nuisance model misspecification, as suggested by the reviewer, which more clearly highlights the advantages of the proposed second–order error structure.
>
>     Regarding “extreme propensity predictions” and “skewed propensity scores,” we explicitly address this through Assumption (A1), which requires the inverse propensity weights to be essentially bounded. This condition rules out arbitrarily large weights and thereby controls instability due to extreme propensities, ensuring that the variance of our estimators remains finite.
>
>     All the above properties are fairly standard in semiparametric causal inference (e.g. [1,2] (Kennedy 2016, 2024)), and we now make these connections explicit in the revision.
>
> 2. **W2. & Q2.** We appreciate this point and agree that quantitative summaries would strengthen the empirical evaluation. While the performance differences between PI, IPW, and AIPW are already visually clear in the figures - AIPW consistently recovers both the shape and magnitude of the true silhouette functions - we have revised the manuscript to include numerical metrics in the Appendix as well. Together with the visualization of average estimate and 1-standard deviation error bands (over 20 repeated simulations) that were provided in our original submission, we now report quantitative estimation bias (in terms of $L_1$ distance to the true effect), and standard deviation summaries. These results, presented in tabular form in Appendix D of the revised manuscript, confirm the visual findings and demonstrate that AIPW achieves substantially improved performance relative to PI and IPW.
>
> 3. **W3.** We thank the reviewer for this helpful suggestion. The improved accuracy of AIPW over PI and IPW in our fully nonparametric setting already demonstrates the practical benefit of its second-order error structure: even when both nuisance functions are estimated flexibly without parametric-rate guarantees, AIPW remains substantially more stable, confirming the theoretical robustness captured in Assumption (A4). That said, we agree that explicit misspecification experiments provide additional insight. In Appendix D.5 of the revised manuscript, we have added extra simulation studies where we deliberately misspecify the propensity score model or the outcome regression. These results further will reinforce the robustness advantages of AIPW in practice.

---

> > ### Author Response · Authors · 2025-11-22
> > **Official Comment by Authors (2/2)**
> >
> > 4. **Q3.** We would like to emphasize that our method is not directly comparable to existing functional causal inference baselines, because the estimand we target is fundamentally different: it captures treatment-induced changes in topological structure, rather than scalar or functional outcomes in Euclidean spaces. In this sense, our framework is built on top of existing functional causal tools rather than competing with them. Most importantly, as emphasized in the main text, our work is, to the best of our knowledge, the first to formally define and estimate causal effects in topological spaces. Consequently, there is currently no existing framework whose target estimand coincides exactly with our proposed topological causal effect.
> >
> >     Although one could apply conventional ATE estimators after somehow collapsing each persistence diagram into an arbitrary finite-dimensional Euclidean summary, such vectorization-based approaches serve only as naive baselines. In contrast, our framework does not treat topological summaries as ad hoc features; rather, it defines the causal estimand directly on the underlying topological space and constructs the silhouette representation as a principled functional embedding that preserves the topological structure. This leads to a well-posed definition of a topological causal effect, together with estimation and inferential procedures tailored specifically to that effect. In the revised text, we have stressed out these points more clearly (lines 186-191, 206-211).
> >
> > We hope that our responses sufficiently resolve your concerns.  If any questions remain, we would be pleased to provide further clarification.
> >
> > [1] Edward H Kennedy. Semiparametric theory and empirical processes in causal inference. Statistical causal inferences and their applications in public health research, pp. 141–167, 2016.
> >
> > [2] Edward H Kennedy. Semiparametric doubly robust targeted double machine learning: a review. Handbook of Statistical Methods for Precision Medicine, pp. 207–236, 2024.

---

### Author Response · Authors · 2025-11-22
**Global comment to all reviewers.**

We are deeply grateful to the reviewers for their careful evaluation and constructive feedback, and we sincerely apologize for the slight delay in our response. Based on the reviewers' valuable comments and feedback, we have made several revisions to our manuscript. We highlight below the major updates incorporated in the newly uploaded version of our manuscript.
- We provide an analysis of estimator bias and variance under explicit nuisance model misspecification (Appendix B).
- We include a subsection dedicated to implementation considerations, such as details of sample splitting and computational cost (Appendix D.1)
- An empirical runtime evaluation for computing the weighted silhouettes is presented in Table 2.
- Quantitative measures of estimator performance - such as estimation bias (measured by the $L_1$ distance to the true effect) and standard deviation summaries - have been added (Table 1).
- We conduct an empirical hypothesis test on the ORBIT dataset and report the results (Appendix D.4).
- Additional experiment results under explicit misspecification of either nuisance model are included in Appendix D.5. Both visual (Figures 13-15) and numerical (Tables 3-5) results are provided.
- Several clarifications have been made throughout the paper.

**All revisions are marked in red** so that they can be easily located. We also note that all references to our paper (such as lines, Sections, etc.) in the responses correspond to the **newly uploaded revised version**.

---

### Meta-Review · Area_Chair_rJvD · 2026-01-06

**Summary:**

This paper proposes a novel framework for causal inference on non-Euclidean outcomes by integrating topological data analysis (TDA) with the potential outcomes paradigm, defining the Topological Average Treatment Effect (TATE) as the expected difference in power-weighted silhouettes of persistence diagrams between treatment groups.

**Reviewer Concerns:**

The idea is considered innovative for causal inference on structural data such as molecular graphs and point clouds (R7QM, ybeB, Zney), theoretically well-developed with asymptotic guarantees, stability bounds, and a consistent hypothesis test (ybeB, Zney, XnWi), and clearly presented with illustrative examples (R7QM, Zney, XnWi). Although some weaknesses remain, including inadequate theoretical rigor (R7QM), limited discussion of computational scalability and hyperparameter sensitivity (ybeB), and room for broader comparison with existing functional causal inference methods (Zney, R7QM), fortunately, the authors have addressed the main issues through clarifications on theoretical guarantees and additional experimental details that would resolve the core concerns raised by reviewers.

**Reviewer Scores:**

R7QM would increase the score as the revision mainly addresses the concerns with the additional theoretical analysis.

ybeB, Zney, and XnWi would keep the score due to the overall positive feedback.

---

### Decision · Program_Chairs · 2026-01-26

Accept (Poster)